# Meaning Representations from Trajectories in Autoregressive Models

**Tian Yu Liu**[†*]
UCLA[1]

**Matthew Trager**[†]
AWS AI Labs[2]

**Alessandro Achille**
AWS AI Labs[2]

**Pramuditha Perera**
AWS AI Labs[2]

**Luca Zancato**
AWS AI Labs[2]

**Stefano Soatto**
AWS AI Labs[2]

## Abstract

We propose to extract meaning representations from autoregressive language models by considering the distribution of all possible trajectories extending an input text. This strategy is prompt-free, does not require fine-tuning, and is applicable to any pre-trained autoregressive model. Moreover, unlike vector-based representations, distribution-based representations can also model asymmetric relations (e.g., direction of logical entailment, hypernym/hyponym relations) by using algebraic operations between likelihood functions. These ideas are grounded in distributional perspectives on semantics and are connected to standard constructions in automata theory, but to our knowledge they have not been applied to modern language models. We empirically show that the representations obtained from large models align well with human annotations, outperform other zero-shot and prompt-free methods on semantic similarity tasks, and can be used to solve more complex entailment and containment tasks that standard embeddings cannot handle. Finally, we extend our method to represent data from different modalities (e.g., image and text) using multimodal autoregressive models. Our code is available at: https://github.com/tianyu139/meaning-as-trajectories

## 1 Introduction

Generative Large Scale Models today are capable of generating remarkably coherent text by iteratively predicting individual tokens. Contrary to encoder-only or encoder-decoder models, however, autoregressive models do not construct explicit representations of sentences: the model's representation of a given input is distributed across layers and attention heads, making it difficult to analyze how the model "understands" and contextualizes language. This lack of transparency and interpretability is a challenge for the responsible deployment of these models.

In this paper, we propose a simple way to explore how *autoregressive models* manipulate text. Whereas standard methods represent sentences by embedding them in a vector space, we propose to represent sentences, or parts of sentences, as the distribution of their possible continuations, or *trajectories*. This can be seen as a practical embodiment of classical distributional approaches to semantics (Boleda, 2020; Sahlgren, 2008), according to which the meaning of linguistic items is tied to the distribution of their usage. It is also related to standard constructions in formal language and automata theory which associate the "behavior" of a prefix with the set of its possible future continuations (Hopcroft et al., 2007).

Prior work has mostly focused on representing sentences via pooling output tokens. For example, BERT-based models (Devlin et al., 2018) include a `[CLS]` token that is meant to capture semantic information. Sentence encoder models such as ST5 (Ni et al., 2021) represent sentences by pooling encoder or decoder outputs. These strategies are most effective after fine-tuning using a contrastive

---

[†]Equal contribution

[*]Work done during an internship at AWS AI Labs.

[1]tianyu@cs.ucla.edu   [2]{mttrager,aachille,pramudi,zancato,soattos}@amazon.com

learning objective (Gao et al., 2021), but this requires additional data and modifies the weights of the original model. In particular, these methods do not faithfully reflect the original model's internal representation of the input sentence, and may be skewed by biases present in the data. Moreover, at the time of writing, the most powerful large language models are based on autoregressive architectures (Touvron et al., 2023; Brown et al., 2020; OpenAI, 2023). For such architectures, similar strategies based on averages of output tokens significantly underperform (Table 1). Instead, *prompt engineering* is usually regarded as the de facto standard to solve semantic tasks without further fine-tuning. For example, Jiang et al. (2023) craft careful prompts to elicit better token representations from autoregressive models. However, such approaches are problematic, since: 1) they can be highly susceptible to the (language dependent) choice of prompt; 2) the answer may not faithfully capture how the model actually interprets the sentence — a model may reply that two sentences are similar, but not necessarily represent them internally in the same way; 3) they do not provide any structured (topological/metric/compositional) semantic space in which sentences are embedded.

Representing the meaning of a sentence as a distribution of trajectories provides a straightforward way of bypassing these limitations. First, the method is general and can be applied to any autoregressive model without assumptions on architecture — or even on the language that it was trained on — and does not require fine-tuning or carefully crafted prompts. Since modern pre-trained large models are very capable of processing text, they should provide strong meaning representations *out-of-the-box*. Second, using representations based on trajectories allows not only measuring semantic distances, but also defining compositional operations on meanings. In particular, we can define Boolean-like operations between meanings, and use them for example to infer the direction of logical entailment between sentences as perceived by the model, or determine hypernym/hyponym relation between words (Figure 2b). Third, our method can be applied without any modification to multimodal autoregressive models that encode images as sequences of tokens, and used to compare the meaning representation of data from different modalities. The main technical challenge of our definition is that the space of all possible continuations of a sentence is too large to be explored directly. We show however that, with an appropriate sampling strategy, 10-20 sampled trajectories for each prompt are sufficient to approximate pairwise distances in semantic space (see Figure 1).

The focus of our work is to propose a general way to extract "canonical" and interpretable meaning representations from pre-trained autoregressive models. Nonetheless, we empirically show that our method achieves competitive results on prompt-free, zero-shot semantic textual similarity (STS) tasks and that the representation we obtain applying our method to the LLaVA (Liu et al., 2023) vision-language model outperforms CLIP embeddings (Radford et al., 2021) on semantic image-image and image-text similarity tasks on the Crisscrossed Captions (Parekh et al., 2020) dataset. We show that the representation we obtain, although based solely on distributions of token sequences, largely agrees with human annotations both on semantic similarity, logical entailment and containment relations. These results support the idea that autoregressive models can represent sentences in a semantically meaningful way even if the representations are not explicit in their activations.

In summary, our main contributions are as follows:

1. We propose a canonical meaning representation for autoregressive models as a distribution over trajectories extending a sentence. Unlike vector space representations, this definition can directly capture asymmetric relations like logical entailments and hypernym/hyponym relations.

2. We show that the representations obtained from modern Large Language Models (LLMs) align well with conventional linguistic meanings: our method achieves competitive performance on Semantic Textual Similarity (STS) benchmarks, outperforming comparable zero-shot and prompt-free baselines using the same architectures.

3. Our method can be extended without any modification to Vision Language Models (VLMs) for the purpose of quantifying semantic image-image and image-text similarity, outperforming even CLIP embeddings (Radford et al., 2021) when applied to LLaVA (Liu et al., 2023).

## 2 RELATED WORK

Our work unifies two lines of work which are highly synergistic yet largely disjoint up until now—the investigation of "meaning" within the internal representations of pre-trained LLMs and the computation of sequence embeddings for semantic comparison tasks.

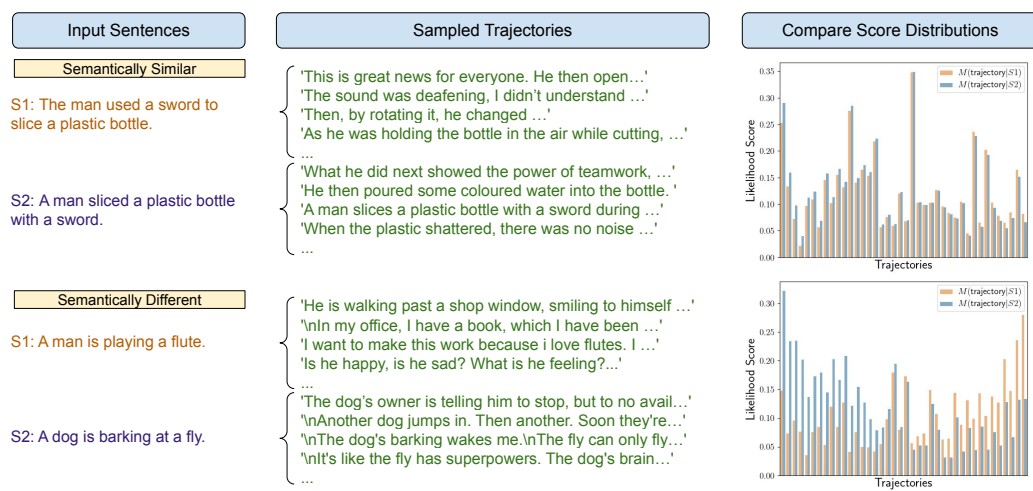

Figure 1: Sentences with similar meanings produce similar score distributions over their continuations (*top*), while sentences with different meanings produce different score distributions over their continuations (*bottom*).

The close relationship between the statistical distribution of linguistic items and their meaning is the basis of the *Distributional Hypothesis* (Harris, 1954). This perspective also draws insight from Wittgenstein's use theory of meaning (Wittgenstein, 1953), commonly sloganized as "meaning is use." In the field of natural language processing (NLP), semantic representations are frequently constructed based on statistical co-occurrences. However, conventional computational approaches such as word2vec (Mikolov et al., 2013), typically involve computing statistics from a text corpus and then constructing vector representations for words or sentences. In contrast, in this work we propose to directly leverage the distribution itself as a fundamental representation of meaning. This is possible since LLMs offer a way to efficiently sample from such distributions, thereby providing an intrinsic notion of meaning from the perspective of the model.

Recently, several authors have argued that models trained on language alone, or more generally on "form," are necessarily incapable of learning and representing conventional meaning. In particular, Bender & Koller (2020) propose a definition of meaning as a relation between language expressions and "communicative intents" which are, by definition, external to the language. Therefore, they conclude that LLMs trained on language expressions *cannot in principle learn meanings*. This leads to characterizing LLMs as "stochastic parrots" (Bender et al., 2021) capable of modeling the statistical form of the language (syntax) but intrinsically incapable of representing meaning (semantics). Merrill et al. (2021) investigate the role of assertions in both code and language, suggesting that ungrounded language models cannot fully emulate semantic representations.

However, semantic structures can be constructed from syntactic ones: For instance, Wu et al. (2023) show that models trained on synthetic languages with "strong transparency" (defined as those where expressions have context-independent denotations) can emulate semantic representations. They suggest, however, that the context-dependency of natural language limits language models from learning the semantic representations within. Using the language of category theory, Bradley et al. (2022) describe a functor between a *syntactic* category of text probabilities and a *semantic* category of meanings. While purely theoretical, their construction is closely related to our distribution-based meaning representation, thus providing further support for our proposed method. Soatto et al. (2023) define meanings in LLMs as equivalence classes of sentences induced by the trained model. This definition generalizes that of Bender & Koller (2020), since "communicative intent" can be latent in the expressions used for training the LLM, which induces partitions the set of complete sentences. But while this characterization is suitable for analyzing the controllability of the model in the corresponding metric space, the resulting meaning representation does not exhibit any obvious compositional structure. Our definition is more general, and provides us with means to compose meaning directly in representation space, unlike all other works. While we do not wish to focus on the high-level aspects of the debate on "meaning" and "understanding" (or lack thereof) in LLMs,

our results provide evidence that autoregressive models actually have rich latent semantic representations within their internal structure.

Encoder-based architectures have traditionally been the main tool for embedding sequences in a common vector space, in which they can be easily compared. Apart from BERT (Devlin et al., 2018) and ST5 (Ni et al., 2021), Sentence-BERT (Reimers & Gurevych, 2019) fine-tunes a modified BERT architecture to improve sentence embeddings. Opitz & Frank (2022) improves the interpretability of Sentence-BERT embeddings while preserving their effectiveness. Zhang et al. (2020) and Gao et al. (2021) propose contrastive fine-tuning objectives to obtain more effective embeddings. In contrast, our method does not require any fine-tuning, hence can faithfully reflect the original model's internal representation of an input string. Prompting is also commonly used to extract embeddings. Jiang et al. (2022) search over prompts to improve the embeddings obtained from BERT. Jiang et al. (2023) propose PromptEOL to summarize sentences as a single word for comparisons. Similar to fine-tuning, prompting alters/biases the meaning of the original string, and further requires sufficient command over the language being used to engineer an effective prompt. The latter also fails to scale with model sizes, generally performing worse on semantic similarity tasks as model size increases.

Most related to our work, Muennighoff (2022) applies decoder-only models for semantic search by computing pairwise conditional likelihood scores between a query and each document in the search database. Our experiments show that this conditional likelihood is insufficient to fully capture relative semantic meaning. Our method is prompt-free, and scales well with model size and human perception of model performances. Unlike prompt-based methods, the meaning space resulting from our method can also be composed to compute more complex relations between strings.

## 3 METHOD

**Preliminaries.** We use $\mathcal{A}$ to denote a finite vocabulary of tokens and $\mathcal{A}^*$ to indicate the set of all variable-length finite sequences of tokens in $\mathcal{A}$. We view a language model as a map $M(\cdot|\cdot) : \mathcal{A}^* \times \mathcal{A}^* \to [0, 1]$ associating a "prompt" sequence $s \in \mathcal{A}^*$ and a possible continuation sequence $t \in \mathcal{A}^*$ with a score $M(t|s) \in [0, 1]$. Intuitively, this score represents the likelihood of the model sampling $t$ as a continuation of $s$. For our experiments, we use as score the inverse perplexity:

$$M(t = (a_1 \ldots a_m)|s) := \prod_{i=1}^{m} P_M(a_i|s\,a_1\ldots a_{i-1})^{1/m}, \tag{1}$$

where $P_M$ is the probability over the next token defined by the model. When $s = \epsilon$ is the empty string, we write $M(t)$ instead of $M(t|\epsilon)$.

**Meaning representation for prompts.** We define the *syntactic meaning representation* of a prompt string $s$ for the model $M$ as the function $M_s := M(-|s) : \mathcal{A}^* \to [0, 1]$. This definition fully captures the way in which the model interprets the string $s$. For example, if $M_s = M_t$, then the prompts $s$ and $t$ are indistinguishable based on their continuations for the model. Note that the function $M_s(t)$ that represents the string $s$ is an infinite dimensional object, since its domain are all finite sequences $t \in A^*$. One of the challenges we will handle later is how to effectively use this representation through sampling.

We remark that we can consider a particular case of eq. (1) where the domain of $M_s$ is restricted only to strings $t \in \mathcal{A}^1$ comprising single tokens, instead of arbitrary length strings. This represents a sentence using the score distribution over the immediate next token, and is a common baseline used to embed sentences with autoregressive models. However, it is easy to see that this is a very incomplete semantic representation: for example, common tokens such as "The" are often the most likely next token regardless of the actual meaning of the prompt. This limitation will be evident in our experimental results.

**Sets of continuations.** The function $M_s : \mathcal{A}^* \to [0, 1]$ essentially represents the meaning of a string as the distribution of trajectories that extend that string. To guide intuitions, it is often useful to consider the more restricted setting where scores are binary $M_s : \mathcal{A}^* \to \{0, 1\}$. This can be interpreted as the characteristic function of the *set* of strings $t$ that are feasible continuations of $s$ according to the model. One advantage of this interpretation is that it makes explicit that meaning representations in our framework are not simple vectors, but rich objects that can be naturally manipulated though set-theoretic operations such as intersections — a fact that we will use later. This simpler setting

also allows a direct connection with *automata theory*: For any language $L \subset \mathcal{A}^*$, the sets of feasible continuations $s^{-1}L := \{t\colon st \in L\}$ of prefixes $s$ can seen as the set of states of a canonical "minimal automaton" accepting the language (Hopcroft et al., 2007). In a similar fashion, the sets $M_s$ of strings accepted by the model prompted with $s$ can be interpreted as a canonical "model of behaviors" for the LLM. We refer to Appendix G for a discussion on these topics.

**Semantic similarity.** Given two prompts $u$ and $v$, we define their *semantic distance* as the distance $d(M_u, M_v)$ between their representation $M_u$ and $M_v$, where $d$ denotes a distance function that can be picked arbitrarily. For our experiments, we use:

$$d(M_u, M_v) = \mathbb{E}_{t \sim \frac{1}{2}(M_u + M_v)} \left| \log M_u(t) - \log M_v(t) \right| \tag{2}$$

$$= \mathbb{E}_{t \sim \frac{1}{2}(M_u + M_v)} \left| \frac{1}{m} \sum_{i=1}^{m} \log \frac{p(a_i | u, a_{<i})}{p(a_i | v, a_{<i})} \right|.$$

This amounts to comparing the expected difference in log-likelihood between the two models on continuations $t \sim \frac{1}{2}(M_u + M_v)$ sampled with equal probability from either prompts. We ablate on other natural choices of distances in Appendix A.2. As noted above, explicitly integrating eq. (2) over all possible trajectories $t$ is not feasible. Rather, we approximate the expectation through Monte Carlo sampling. More precisely, we sample $n$ trajectories $T_u = \{t_i^u\}_{i=1}^{n}$ for the prompt $u$, where $t_i^u \sim M_u$, and $n$ trajectories $T_v$ for the prompt $v$, each of length up to a fixed hyperparameter $m$. We then approximate eq. (2) as:

$$d(M_u, M_v) \approx \frac{1}{2n} \sum_{t \in T_u \sqcup T_v} \left| \log M_u(t) - \log M_v(t) \right|$$

The steps we follow are detailed in Algorithm 1. More sophisticated approaches for approximating the distance could be explored in future work.

A related baseline for comparing the similarity of two sentences $u$ and $v$ is the likelihood of their concatenation, $M(uv)$ or $M(v|u)$. However, perplexity-based measures are known to be unreliable when directly used to compare different sentences, even when the sentences have the same length (Wang et al., 2022; Meister & Cotterell, 2021). Moreover, the fact that $v$ is a likely continuation of $u$ does not necessarily imply that $u$ and $v$ have the same meaning. Our method circumvents these problems: rather than computing $M(v|u)$, we compare the values of $M_u(t) = M(t|u)$ and $M_v(t) = M(t|v)$ on a common set of continuations $t \in T_u \sqcup T_v$. This strategy is arguably more natural and also, as our experiments will demonstrate, much more effective. While our notions of semantic similarity are defined from the perspective of language models, our experiments in Section 4 suggest that they increasingly align with that of human annotators as model size and training data increases, and vastly outperform that of next-token/likelihood baselines.

---

**Algorithm 1** Similarity in Meaning Space

---

**Require:** Model $M$, Strings $u$ and $v$, num. trajectories $n$, max trajectory length $m$, distance $d$
    $T_u \leftarrow$ Sample $n$ trajectories from $u$ up to [EOS] or length $m$, whichever occurs sooner
    $T_v \leftarrow$ Sample $n$ trajectories from $v$ up to [EOS] or length $m$, whichever occurs sooner
    Initialize $M_u = M_v = \emptyset$
    **for** $t = a_1 \ldots a_{m_t} \in T_u \sqcup T_v$ **do**                ▷ Compute trajectory likelihood
        $M_u[t] \leftarrow \prod_{i=1}^{m_t} P_M(a_i | u\, a_1 \ldots a_{i-1})^{1/m_t}$
        $M_v[t] \leftarrow \prod_{i=1}^{m_t} P_M(a_i | v\, a_1 \ldots a_{i-1})^{1/m_t}$
    **end for**
    **return** $d(M_u, M_v)$                                   ▷ Return similarity score

---

**Containments of semantic representations.** Our representations $M_u$ belong to the space of functions $[0,1]^{\mathcal{A}^*}$, which we can view as the "meaning space" for the vocabulary $\mathcal{A}$. Note that this space has a natural *partial order*: given $M, N \in [0,1]^{\mathcal{A}^*}$, we say that $M < N$ if $t \in \mathcal{A}^*$ we have $M(t) < N(t)$, which intuitively means that any feasible sentence for $M$ is also feasible for $N$. More generally, we can define operations of *meet* and *join* as $M \wedge N := \min(M, N)$ and $M \vee N := \max(M, N)$, respectively. These Boolean-like operations on meanings can be used to investigate more complex (even asymmetric) meanings relationships, in addition to similarity.

These definitions require using unnormalized scores, which is why we consider $[0,1]^{\mathcal{A}^*}$ instead of the set of probabilities over $\mathcal{A}^*$ as our meaning space. Note that, in contrast, traditional vector-space embeddings are ill-suited for representing such relationships.

In our experiments, we explore how this sort of (syntactic) meaning containment is related to *entailment* ($\Rightarrow$) in the conventional sense. As we discuss in Appendix F, given two sentences $u$ and $v$ such that $u \Rightarrow v$, the relation $M_v < M_u$ is "more true" than $M_u < M_v$. Note that, for our particular score representation in eq. (1), neither $M_v < M_u$ nor $M_v > M_u$ can hold exactly; however we can quantify how far they are from being true. Based on this, we define the *Entailment Test:* If $d(M_u \wedge M_v, M_v) < d(M_u \wedge M_v, M_u)$, then $u \Rightarrow v$; otherwise, $v \Rightarrow u$.

**Semantic representation for substrings.** The meaning representation $M_s$ for a string considered until now assumes that $s$ is used as a prompt, i.e., as a prefix within a longer string. We can also modify our definition to account for strings in any position, and in particular to words. Specifically, for any string $u$, we consider a meaning representation $\overline{M}_u : \mathcal{A}^* \times \mathcal{A}^* \to [0,1]$ defined by:

$$\overline{M}_u(s,t) := M(s\,u\,t).$$

Intuitively, the meaning of a word/string is the likelihood function of it appearing in between all "contexts" $(s,t)$ — a very natural idea in distributional semantics, resembling for example the skip-gram model used in word2vec (Mikolov et al., 2013).

Using this representation, we can define partial ordering of meanings in the same way considered above for prompts. However, unlike the previous setting, sampling the support of $\overline{M}_u$ or $\overline{M}_v$ (contexts that contain $u$ and $v$) is not trivial, since LLMs can only sample "forward" trajectories. In practice, we circumvent this issue by using a text corpus, WikiText (Merity et al., 2016), to retrieve, rather than sample, paragraphs containing the given word to use as context. In Section 4, our experiments show that the partial ordering in semantic space aligns quite well with "meaning containment" in natural language, i.e., with hyponym/hypernym relations defined in WordNet (Miller, 1995). Specifically, if $v$ is a hyponym of $u$, then it is natural to expect that $\overline{M}_v < \overline{M}_u$ (see Appendix F for a justification). Thus, given two words $(u,v)$ between which a hyponym relation exists, we define the *Hyponym Test*: If $d(\overline{M}_u \wedge \overline{M}_v, \overline{M}_v) < d(\overline{M}_u \wedge \overline{M}_v, \overline{M}_u)$, then $v$ is a hyponym of $u$; otherwise, $u$ is a hyponym of $v$. We refer to Algorithm 2 in the Appendix for full details.

**Semantic similarity for different modalities.** The meaning representations we consider are applicable to any model that assigns likelihoods to sequences of tokens. In particular, they can be applied without modification to multimodal autoregressive models which accept both image and text prompts. In Section 4, we show how meaning representations obtained from the multimodal model LLaVA (Liu et al., 2023) can effectively compute semantic image-text and image-image distances.

## 4 EXPERIMENTS

**Implementation details.** Apart from adding a full stop (".") at the end of each sequence that does not already end with a punctuation to complete the sentence, we evaluate each dataset verbatim (in Table 6 in the Appendix, we show results obtained without this step). For our baseline methods, we report the best result with or without adding a full stop, to ensure fair comparison. For experiments on LLaVA (Liu et al., 2023), we use the default query format to structure the input data. We do not apply any additional prompts/formatting for all other models unless otherwise mentioned. We use eq. (2) as our distance function. We report results using other metrics/divergences in the Appendix. We use multinomial sampling for all experiments on our method with sampling temperature $\lambda = 1.0$. We set $n = 20$ and $m = 20$ for sampling trajectories, based on ablations in Appendix A. Distance metric and hyperparameter choices for semantic similarity are based on a search using the validation set of the STS-B dataset, and are then fixed when evaluating on all test datasets.

**Evaluation procedure.** We evaluate our method on the following tasks:

*Semantic Textual Similarity (STS) (Agirre et al., 2012; 2013; 2014; 2015; 2016; Cer et al., 2017)*: The STS dataset scores how similar two pieces of texts are. We use the Spearman coefficient (scaled by $100\times$) to evaluate correlation with the human-annotated similarity scores.

*Stanford Natural Language Inference (SNLI) (Bowman et al., 2015):* SNLI labels pairs of strings based on the categories {entailment, neutral, contradiction}. The latter two are symmetric and can

Table 1: Comparison with other prompt-free and zero-shot methods on Semantic Textual Similarity benchmarks. $*$; $\dagger$ indicate results taken from Ni et al. (2021); Gao et al. (2021) respectively. Our method outperforms all baselines, and even encoder-based methods like ST5-Enc-mean (11B). As model size scales, our method approaches the paragon of contrastive-trained models, even though the models we use have been trained only on unsupervised next-token prediction.

| | STS-B | STS12 | STS13 | STS14 | STS15 | STS16 | SICK-R | Avg |
|---|---|---|---|---|---|---|---|---|
| *Paragon: Contrastive-Trained Models* | | | | | | | | |
| CLIP-ViTL14 (Radford et al., 2021) | 65.5 | 67.7 | 68.5 | 58.0 | 67.1 | 73.6 | 68.6 | 67.0 |
| IS-BERT † (Zhang et al., 2020) | 56.8 | 69.2 | 61.2 | 75.2 | 70.2 | 69.2 | 64.3 | 66.6 |
| SimCSE-BERT † (Gao et al., 2021) | 68.4 | 82.4 | 74.4 | 80.9 | 78.6 | 76.9 | 72.2 | **76.3** |
| *Zero-Shot Encoder-based Models* | | | | | | | | |
| BERT-CLS$^*$ (Devlin et al., 2018) | 16.5 | 20.2 | 30.0 | 20.1 | 36.9 | 38.1 | 42.6 | 29.2 |
| BERT-mean$^*$ (Devlin et al., 2018) | 45.4 | 38.8 | 58.0 | 58.0 | 63.1 | 61.1 | 58.4 | 54.8 |
| BERT Large-mean$^*$ (Devlin et al., 2018) | 47.0 | 27.7 | 55.8 | 44.5 | 51.7 | 61.9 | 53.9 | 48.9 |
| RoBERTa Large-mean$^*$ (Liu et al., 2019) | 50.6 | 33.6 | 57.2 | 45.7 | 63.0 | 61.2 | 58.4 | 52.8 |
| ST5-Enc-first (Base)$^*$ (Ni et al., 2021) | 16.7 | 17.5 | 6.3 | -20.7 | 2.3 | 21.9 | 28.6 | 10.4 |
| ST5-EncDec-first (Base)$^*$ (Ni et al., 2021) | 9.4 | 10.9 | 29.6 | 14.9 | 28.9 | 30.6 | 39.3 | 23.4 |
| ST5-Enc-mean (Large)$^*$ (Ni et al., 2021) | 56.3 | 28.0 | 52.6 | 41.4 | 61.3 | 63.6 | 59.5 | 51.8 |
| ST5-Enc-mean (11B)$^*$ (Ni et al., 2021) | 62.8 | 35.0 | 60.2 | 47.6 | 66.4 | 70.6 | 63.6 | 58.0 |
| *Autoregressive Model Baselines: Falcon-7B* | | | | | | | | |
| Cross Encoder (Muennighoff, 2022) | 46.7 | 25.1 | 53.9 | 41.9 | 53.7 | 54.2 | 57.2 | 47.5 |
| Joint Likelihood | 38.1 | 6.0 | 40.8 | 32.7 | 33.7 | 35.7 | 47.6 | 33.5 |
| Last token | 23.1 | 27.0 | 20.1 | 8.5 | 18.7 | 18.3 | 40.8 | 22.4 |
| Mean token | 18.8 | 18.0 | 25.9 | 18.5 | 25.8 | 27.5 | 37.3 | 24.5 |
| *Autoregressive Models (Ours)* | | | | | | | | |
| Ours (GPT-2) | 55.2 | 39.9 | 42.6 | 30.5 | 52.4 | 62.7 | 62.0 | 49.3 |
| Ours (GPT-2-XL) | 62.1 | 43.6 | 54.8 | 37.7 | 61.3 | 68.2 | 68.4 | 56.5 |
| Ours (Falcon-7B) | 67.7 | 56.3 | 66.5 | 53.0 | 67.4 | 75.5 | 73.5 | 65.7 |
| Ours (LLaMA-13B) | 70.6 | 52.5 | 65.9 | 53.2 | 67.8 | 74.1 | 73.0 | 65.3 |
| Ours (LLaMA-33B) | 71.5 | 52.5 | 70.6 | 54.6 | 69.1 | 75.2 | 73.0 | **66.6** |

be quantified via similarity. To evaluate our method's ability to compute asymmetric relationships, we restrict SNLI to only pairs of sentences labelled with the "entailment" relation. We express this as a binary classification task to determine the direction of entailment, i.e., given pair $(u, v)$, we wish to determine if $u \Rightarrow v$, or $v \Rightarrow u$. We term this resultant task *SNLI-Entailment*.

*WordNet (Miller, 1995):* WordNet establishes a hierarchy among English words through semantics-based hypernym/hyponym relations. We sample branches from the WordNet hierarchy (see Appendix C.1), and recover their pairwise relations using operations in syntactic meaning space.

*Crisscrossed Captions (CxC) (Parekh et al., 2020):* CxC extends MS-COCO (Lin et al., 2014) with human-labelled semantic similarity scores ranging from 0-5 for image-image, caption-caption, and image-caption pairs. Since most scores are close to 5 (e.g. original image-caption pairs from COCO) for which ranking comparisons would be vacuous, we subsample a balanced subset of 1000 pairs each from the image-image (CxC-SIS) and image-caption (CxC-SITS) dataset for our experiments.

**Semantic similarity.** Our main baselines for comparison are methods which are 1) zero-shot, and 2) prompt-free. As such, we compare our method as presented in Algorithm 1 against encoder-based models, and the following baselines for autoregressive models given a pair of strings $(u, v)$:

1. *Conditional Likelihood / Cross-Encoder* (Muennighoff, 2022): computes $M(u|v)$.

2. *Joint Likelihood*: measures the likelihood of the concatenation of $u$ and $v$, $M(uv) = M(uv|\epsilon)$ where $\epsilon$ is begin-of-sentence token [BOS], normalized by number of tokens. If [BOS] is not supported by the model, we use the $M(u\, v_n \ldots v_2|v_1)$ instead where $v = (v_n \ldots v_1)$.

3. *(Last/Mean) Token:* we represent $u$ and $v$ using the model's output distribution for the next token immediately following the sentence (last) or the average next-token predictions over the input sentence (mean), and produce a similarity score via cosine similarity.

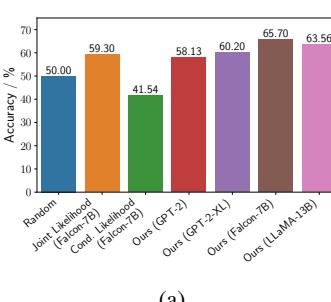
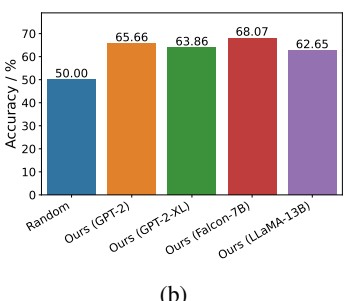
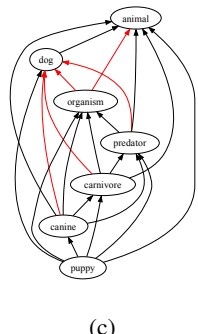

|     (a)     |     (b)     |     (c)     |

Figure 2: **(a) Accuracy in inferring the entailment direction.** On SNLI-Entailment, our method outperforms existing baselines applied to the best model, Falcon-7B, showing that our notion of meaning containment aligns with that of natural language when quantifying "entailment" relationships between statements. **(b) Accuracy in inferring hypernym/hyponym direction.** Our method performs significantly better than chance on WordNet hyponym/hypernym prediction. **(c) Visualization of word hierarchy recovered by our method** on a subset of words using Falcon-7B (red indicates predictions that differ from the WordNet ground-truth).

On the Semantic Textual Similarity benchmark, Table 1 shows that our method uniformly outperforms all baselines on one of the best autoregressive models, Falcon-7B, by a minimum relative improvement of $38.3\%$. Even when applied to GPT-2, a much smaller model, our method improves over Falcon-7B baselines by $3.8\%$. While our method is expectedly outperformed by models which are explicitly fine-tuned on contrastive-learning objectives, such as SimCSE (Gao et al., 2021), it performs comparably to CLIP (Radford et al., 2021), and when applied to Falcon-7B and LLaMA-33B respectively, significantly outperforms the best zero-shot encoder-based model (ST5-Enc-mean 11B) by a relative margin of $13.3\%$ and $14.8\%$. We achieve this without any fine-tuning or prompting. We further highlight that our results do not rely on any human-annotated data or contrastive pairs, since the models we use have been trained only on unsupervised next-token prediction.

Lastly, our method shows an improvement in performance that correlates with model size, suggesting that further performance gains could be obtained as larger/better autoregressive models are used. Our results also suggest that the proposed method can be used to evaluate pre-trained models in a zero-shot manner without requiring instruction-tuning or RLHF, since their alignment with human labelers seems to correlate with human perception of how good a model is.

**Entailment via meaning containment.** We show accuracies obtained on SNLI-Entailment in Figure 2a when applying the Entailment Test described in Section 3. We compare against Cond. Likelihood ($u \Rightarrow v$ if $M(v|u) > M(u|v)$, else $v \Rightarrow u$) and Joint Likelihood ($u \Rightarrow v$ if $M(uv) > M(vu)$) on the best performing model, Falcon-7B. Our results show that the trajectories sampled from all LLMs that we tested align with the assumptions of the Entailment Test with significantly high probability, outperforming both random and likelihood baselines by $15.7\%$ and $9.3\%$ respectively.

**Meaning containment of individual words.** We apply the above-defined Hyponym Test to recover hypernym/hyponym relations from WordNet. Our results in Figure 2b and Figure 2c show that the Hyponym Test is mostly able to recover semantic containment relations between words, with an absolute improvement of $12.7\%$ to $18.1\%$ over the random baseline, depending on the model. Note that our computation of the hierarchy is based entirely on pairwise comparisons and does not explicitly enforce the transitivity of containments; however, transitivity is almost always already satisfied by the predictions of our method (i.e., the recovered hierarchy is an acyclic graph). We present more qualitative examples in Appendix D.2.

**Vision-language experiments.** Our method can be applied without any modification to models that accept multimodal token sequences. In Table 2, we apply our method to CxC (Parekh et al., 2020) using the vision-language model LLaVA (Liu et al., 2023) to show that we can measure semantic distances between not only text, but also between image-image (CxC-SIS) and image-text (CxC-SITS) pairs that align with that of human annotators. Our method outperforms all decoder-only baselines on both SIS and SITS. On SIS, our method even outperforms CLIP (Radford et al., 2021) which is trained explicitly on a contrastive image-text objective.

Table 2: Image-Image Similarity and Image-Text (Caption) Similarity on balanced subsets of CxC-SIS and CxC-SITS respectively. Even without any prompts, our method outperforms all zero-shot baselines on both modalities. The performance on the image-text similarity can be further boosted with an alignment prompt, allowing our method to outperform even CLIP which is explicitly trained with a contrastive objective to output aligned image-text embeddings. For CLIP (Vision), we use image embeddings prior to projection onto text embedding space.

| Architecture | Method | CxC-SIS | CxC-SITS | Average |
|---|---|---|---|---|
| CLIP (Radford et al., 2021) | CLIP-ViTL/14 | 66.33 | 64.25 | 65.29 |
| | CLIP-ViTB/16 | 66.95 | **64.60** | **65.78** |
| | CLIP-ViTL/14 (Vision) | 71.45 | - | - |
| | CLIP-ViTB/16 (Vision) | 72.08 | - | - |
| LLaVA (Liu et al., 2023) | Cond. Likelihood | - | 29.46 | - |
| | Mean Token | 32.76 | -0.52 | 16.12 |
| | Last Token | 26.91 | 2.43 | 14.67 |
| | **Ours** | **81.47** | **57.14** | **69.31** |
| LLaVA (Liu et al., 2023) w/ Alignment Prompt | Mean Token (Prompt) | 32.76 | -0.07 | 16.35 |
| | Last Token (Prompt) | 26.91 | 6.21 | 16.56 |
| | **Ours (Prompt)** | **81.47** | **67.63** | **74.55** |

We highlight that while the "Cond. Likelihood" baseline on SITS should most directly capture $M(\text{caption}|\text{image})$, our experiments show that it fares poorly compared to our method. We hypothesize that this results from the limitations of perplexity. For instance, likelihood scores are directly compared across captions of various lengths, for which length normalization does not sufficiently mitigate the bias towards shorter sentences (Wang et al., 2022). Our method avoids this issue entirely by construction, since we compare distributions across the same set of trajectories. We can optionally make use of "alignment prompts" to ensure that the trajectories from image and text modalities are more similar. This improves the resulting performance on the CxC-SITS task, outperforming the CLIP paragon by 13.3% (Table 2). We discuss this in Appendix E.2.

## 5 CONCLUSIONS

We proposed a strategy to investigate how autoregressive language models interpret text. By identifying "meaning" — from the perspective of the model — with score distributions over text continuations, we can compare the meaning of arbitrary strings. This notion of meaning correlates with that of human annotators, outperforming comparable zero-shot and prompt-free baselines on semantic textual similarity tasks using the same architectures. We further defined composition operators on meanings and showed how autoregressive language models can be used to quantify entailment between sentence pairs and hyponym relations between individual words. Our method can further be applied without modification to autoregressive vision-language architectures to define and compute meaning representations of images, outperforming even CLIP on semantic image similarity tasks.

A key limitation of our approach is its computational cost compared to embedding methods that require only a single forward pass. However, our ablations in Appendix A show that that using 10-20 trajectories of 10-20 tokens each is sufficient to achieve most of the performance gain compared to using more trajectories or tokens. We also note that both the sampling and score evaluation processes can be easily parallelized. Our approach in its current form is also not computationally efficient for semantic search, since the computation of pairwise similarities between queries and database elements is performed using a different set of trajectories for each new query. We explore ways to mitigate this in Appendix A.4 and the potential performance trade-offs that they incur.

Our method is intentionally prompt-free, as our goal in this work was to define the most canonical meaning representation of a string for a given model. Nevertheless, our experiments in Appendix E strongly suggest that designing appropriate "alignment" prompts could further significantly improve quantitative results on semantic similarity tasks. Our method can also be generalized to compare semantic distances between autoregressive models from the same family of architectures sharing a common vocabulary, since their meaning representations belong to the same space. We leave these directions for future work.

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

# Supplementary Material

## A  ABLATION STUDIES

In this section, we present ablation studies on how trajectories are sampled in Appendix A.1, choice of distance function in Appendix A.2, completing "incomplete" sentences with a single full stop in Appendix A.3, and discuss extensions to perform computationally efficient semantic search in Appendix A.4.

### A.1  ABLATION ON TRAJECTORIES

We present ablations on Algorithm 1 to investigate the impact of (1) number of trajectories (2) length of trajectories and (3) sampling temperature. All experiments are done on the validation set of STS-B instead of test set to avoid over-fitting results to the test set.

Figure 3(a) shows that performance on evaluating semantic similarity increases with both number and length of trajectories sampled, at the cost of computational time. We use $n = m = 20$ for all of our main experiments, which is sufficient to yield most of the performance. We also ablate of sampling temperature $\lambda$ in Figure 3(b), where we show that sampling trajectories that are either too diverse or lack diversity (as measured by $\lambda$) tends to harm performance. Instead, the standard temperature value $\lambda = 1.0$ yields the best results.

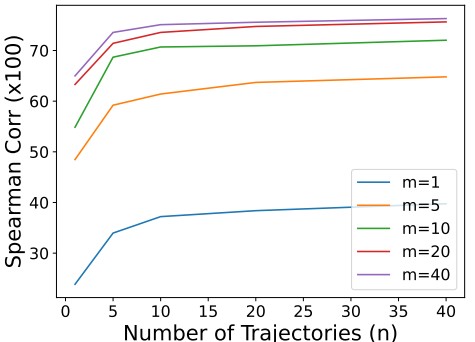 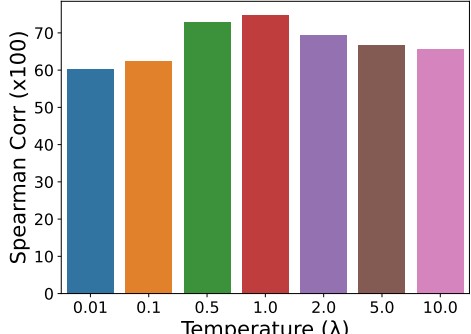

Figure 3: Ablation over maximum length (M), number (N) of trajectories, and sampling temperature ($\lambda$) on STS-B validation dataset using the Falcon-7B model. While only a small number of short trajectories is sufficient to yield good results, performance on semantic similarity generally increases with both number and length of trajectories. Too much diversity and lack of diversity in the sampled trajectories both harm performance, as shown by higher and lower values of $\lambda$ respectively.

### A.2  CHOICE OF DISTANCE FUNCTION

We further ablate over the choice of distance function in Table 3. For distance functions on probability spaces (Hellinger, Total Variation, Symmetric KL-Divergence), we normalize the scores $M_u$ using

$$M_u^{norm}(t) := \frac{M_u(t)^\tau}{\sum_{t' \in \mathcal{A}^*} M_u(t')^\tau} \tag{3}$$

to convert them into a probability distribution summing to 1. We use $\tau = 0.5$ which we experimentally determined to perform best. We compare against our choice of distance function as defined in eq. (2), and a modified version that uses L2 instead of L1 loss, which we refer to as Log-L1 and Log-L2 respectively. We show that most choices of distance functions (Symmetric KL Divergence, Hellinger distance, Log-L2, Log-L1) work reasonably well for computing the semantic distance between strings. We chose Log-L1 in our main experiments, which performs best.

Table 3: Ablation over the choice of distance function on LLaMA-13B and LLaVA. For Hellinger distance, Total Variation (TV), and Symmetric KL-Divergence, we set $\tau = 0.5$ as in eq. (3). Here we use the prompt-aligned version of our method for SITS.

| Metric | STS-B | STS-12 | STS-13 | STS-14 | STS-15 | STS-16 | SNLI | SIS | SITS |
|--------|-------|--------|--------|--------|--------|--------|------|-----|------|
| Hellinger | 69.7 | 53.0 | 65.5 | 50.7 | 65.6 | 71.2 | 65.8 | 81.3 | 67.6 |
| TV | 51.2 | 40.9 | 47.8 | 32.8 | 42.4 | 50.9 | 64.1 | 78.5 | 62.4 |
| Sym-KL | 69.7 | 52.9 | 65.4 | 50.6 | 65.5 | 71.1 | 65.9 | 80.9 | 67.6 |
| Log-L1 | 70.6 | 52.5 | 65.9 | 53.2 | 67.8 | 74.1 | 63.6 | 81.0 | 67.6 |
| Log-L2 | 69.2 | 51.4 | 64.1 | 48.2 | 65.7 | 71.6 | 65.4 | 80.8 | 67.0 |

### A.3 COMPLETING THE SENTENCE WITH FULL STOP

According to our definitions, the meaning representations of complete and incomplete sentences differ, since the distributions over their trajectories are likely to be very different. To see this, consider the following pair of semantically similar sentences that differ by the last punctuation: "The dog ate the bone" and "The dog ate the bone.". The continuations of the latter are likely to start with a capital letter, but this does not hold for the former. Hence, our method is likely to attribute larger distances between these two sentences than humans. In Table 4, we show that by ensuring all sentences we compare are complete, by appending a full stop when necessary, the similarity scores computed between sentences align better with that of human annotators. We also show that completing the sentence can occasionally improve results for certain baselines as well.

### A.4 EXTENSION TO SEMANTIC SEARCH

We note that Algorithm 1 is computationally expensive for semantic search, where we wish to retrieve the most similar sample in a search database $\mathcal{D}$ given a query $q$, since it requires multiple sampling and forward pass operations for each pairwise comparison $d(q, s)$ for all $s \in \mathcal{D}$. As such, an inference cost of $\mathcal{O}(|\mathcal{D}|)$ is incurred each time a new query is received. This holds true even for previously proposed methods for semantic search using decoder-only models, e.g., (Muennighoff, 2022). Instead, if there exists a fixed set of trajectories $T_{\mathcal{D}}$ for the search database $\mathcal{D}$ that can be used instead of $\mathcal{A}^*$ in eq. (1), then $M_s$ for each item $s \in \mathcal{D}$ can be pre-computed beforehand, incurring a one-time cost of $\mathcal{O}(|\mathcal{D}|)$. Hence, for each subsequent query, we only need to incur an $\mathcal{O}(1)$ inference cost to compute $M_q$ on $T_{\mathcal{D}}$. This can be compared against the pre-computed embeddings in $\mathcal{D}$ using standard distance functions such as L1. We present a proof-of-concept experiment on the STS-B validation dataset to observe the trade-off in performance that this incurs in Table 5, where we obtain $T_{\mathcal{D}}$ by naively selecting $n$ examples from the dataset uniformly at random, from each of which we generate a single trajectory. Nevertheless, our preliminary results demonstrate that it is indeed possible to achieve satisfactory performance using fixed sets of trajectories. We leave investigating more sophisticated methods to construct $T_{\mathcal{D}}$ for future work.

## B FURTHER BASELINES

We provided baseline comparisons in Table 1 of the main body of the paper against one of the best model tested, Falcon-7B. In Table 6, we provide additional baseline results for several other autoregressive architectures used.

## C ADDITIONAL IMPLEMENTATION DETAILS

We use the base GPT-2, GPT-2-XL (Radford et al., 2019), LLaMA-13B (Touvron et al., 2023) and Falcon-7B (Almazrouei et al., 2023) for experiments on models trained with unsupervised pre-training objectives. We use Vicuna-13B (Chiang et al., 2023) and StableVicuna-13B[1] as the instruction-tuned version and the RLHF-trained (reinforcement learning with human feedback) ver-

---

[1]https://huggingface.co/CarperAI/stable-vicuna-13b-delta

Table 4: Ablation over adding a full stop (FS) to incomplete sentences. Based on our definitions, meaning representations of complete and incomplete sentences differ. We show that the meaning similarities of complete sentences align better with that of human annotators.

| Method | FS? | STS-B | STS12 | STS13 | STS14 | STS15 | STS16 | SICK-R | Avg |
|---|---|---|---|---|---|---|---|---|---|
| **Baselines (Falcon-7B)** | | | | | | | | | |
| Cond. Likelihod | ✗ | 46.0 | 22.3 | 52.5 | 41.7 | 46.9 | 51.8 | 54.3 | 45.1 |
| Joint Likelihood | ✗ | 38.3 | 4.5 | 38.3 | 32.4 | 28.3 | 34.4 | 43.3 | 31.4 |
| Last token | ✗ | 24.9 | 18.9 | 13.6 | 4.2 | 4.7 | 18.5 | 34.1 | 17.0 |
| Mean token | ✗ | 18.8 | 18.0 | 25.9 | 18.5 | 25.8 | 27.5 | 37.3 | 24.5 |
| Cond. Likelihood | ✓ | 46.7 | 25.1 | 53.9 | 41.9 | 53.7 | 54.2 | 57.2 | 47.5 |
| Joint Likelihood | ✓ | 38.1 | 6.0 | 40.8 | 32.7 | 33.7 | 35.7 | 47.6 | 33.5 |
| Last token | ✓ | 23.1 | 27.0 | 20.1 | 8.5 | 18.7 | 18.3 | 40.8 | 22.4 |
| Mean token | ✓ | 18.1 | 21.7 | 25.4 | 16.9 | 26.3 | 26.6 | 33.9 | 24.1 |
| **Baselines (LLaMA-13B)** | | | | | | | | | |
| Cond. Likelihood | ✗ | 41.9 | 19.8 | 54.6 | 40.1 | 54.6 | 52.2 | 55.0 | 45.5 |
| Joint Likelihood | ✗ | 36.6 | -0.6 | 34.8 | 27.8 | 28.2 | 32.6 | 43.2 | 28.9 |
| Last token | ✗ | 18.8 | 15.9 | 18.2 | 5.6 | 2.3 | 9.9 | 35.6 | 15.2 |
| Mean token | ✗ | 28.0 | 22.0 | 27.5 | 19.6 | 30.8 | 35.8 | 43.7 | 29.6 |
| Cond. Likelihood | ✓ | 44.3 | 20.8 | 51.8 | 38.6 | 56.0 | 50.9 | 56.7 | 45.6 |
| Joint Likelihood | ✓ | 36.7 | 1.1 | 35.0 | 27.7 | 33.0 | 32.4 | 48.0 | 30.6 |
| Last Token | ✓ | 18.2 | 24.2 | 29.0 | 16.8 | 21.9 | 10.2 | 40.8 | 23.0 |
| Mean Token | ✓ | 28.8 | 25.2 | 30.2 | 20.2 | 31.5 | 35.1 | 45.0 | 30.9 |
| **Ours** | | | | | | | | | |
| Ours (GPT-2) | ✗ | 48.3 | 28.7 | 39.7 | 23.8 | 35.7 | 60.0 | 56.6 | 41.8 |
| Ours (GPT-2-XL) | ✗ | 56.8 | 32.5 | 49.5 | 29.1 | 45.2 | 66.0 | 63.1 | 48.9 |
| Ours (Falcon-7B) | ✗ | 67.2 | 44.0 | 62.1 | 44.7 | 57.5 | 76.1 | 69.3 | 60.1 |
| Ours (LLaMA-13B) | ✗ | 66.9 | 39.9 | 61.2 | 45.0 | 56.4 | 74.4 | 68.7 | 58.9 |
| Ours (GPT-2) | ✓ | 55.2 | 39.9 | 42.6 | 30.5 | 52.4 | 62.7 | 62.0 | 49.3 |
| Ours (GPT-2-XL) | ✓ | 62.1 | 43.6 | 54.8 | 37.7 | 61.3 | 68.2 | 68.4 | 56.6 |
| Ours (Falcon-7B) | ✓ | 67.7 | 56.3 | 66.5 | 53.0 | 67.4 | 75.5 | 73.5 | 65.7 |
| Ours (LLaMA-13B) | ✓ | 70.6 | 52.5 | 65.9 | 53.2 | 67.8 | 74.1 | 73.0 | 65.3 |

Table 5: Trade-off in performance on STS-B validation set from using a fixed set of trajectories ($m = 20$) for all pairwise distance comparisons.

| Method | Spearman Corr (x100) |
|---|---|
| Ours (Falcon-7B) | 74.74 |
| - Fixed Traj ($n = 20$) | 49.41 |
| - Fixed Traj ($n = 40$) | 53.29 |

sion of LLaMA-13B respectively. We use LLaVA[1] (Liu et al., 2023) for our multimodal experiments, which is trained to accept both image and text inputs.

Technically, computing distances with Equation (2) when compositional terms are involved would require sampling trajectories from the composed distributions. In particular, evaluating $d(M_u \wedge M_v, M_u)$ in the Entailment test would require sampling trajectories $T_{u \wedge v}$ from the composed distribution $M_u \wedge M_v$, then approximating Equation (2) with the set of trajectories $T_{u \wedge v} \sqcup T_u$. For the sake of simplicity and computational efficiency, we instead compute $M_u \wedge M_v$ over the set of trajectories $T_u \sqcup T_v$ sampled from $u$ and $v$, which we empirically found to be similarly effective when applied to downstream tasks.

---

[1]https://huggingface.co/liuhaotian/llava-v1-0719-336px-lora-merge-vicuna-13b-v1.3

Table 6: Comparison of our method against baselines (best among with/without fullstop) for each model architecture on STS tasks.

| Model | Method | STS-B | STS12 | STS13 | STS14 | STS15 | STS16 | SICK-R | Avg |
|---|---|---|---|---|---|---|---|---|---|
| GPT-2 | Cond. Likelihood | 37.9 | 28.6 | 39.1 | 34.3 | 50.5 | 47.1 | 53.0 | 41.5 |
| | Joint Likelihood | 27.9 | 18.4 | 22.0 | 23.1 | 32.8 | 27.2 | 44.0 | 27.9 |
| | Last token | 27.7 | 8.4 | 23.0 | 10.5 | 31.0 | 26.6 | 41.9 | 24.1 |
| | Mean token | 20.4 | 17.0 | 22.0 | 17.6 | 36.2 | 31.5 | 38.4 | 26.2 |
| | Ours | 55.2 | 39.9 | 42.6 | 30.5 | 52.4 | 62.7 | 62.0 | **49.3** |
| GPT-2-XL | Cond. Likelihood | 40.3 | 23.8 | 43.2 | 33.6 | 51.3 | 48.7 | 55.0 | 42.3 |
| | Joint Likelihood | 31.4 | 13.3 | 29.3 | 23.8 | 35.2 | 28.6 | 46.0 | 29.7 |
| | Last token | 24.1 | -5.9 | 20.9 | 5.0 | 25.6 | 21.4 | 40.8 | 18.8 |
| | Mean token | 21.1 | 13.0 | 28.0 | 16.7 | 34.9 | 33.0 | 37.8 | 26.4 |
| | Ours | 62.1 | 43.6 | 54.8 | 37.7 | 61.3 | 68.2 | 68.4 | **56.6** |
| Falcon-7B | Cond. Likelihood | 46.7 | 25.1 | 53.9 | 41.9 | 53.7 | 54.2 | 57.2 | 47.5 |
| | Joint Likelihood | 38.1 | 6.0 | 40.8 | 32.7 | 33.7 | 35.7 | 47.6 | 33.5 |
| | Last token | 23.1 | 27.0 | 20.1 | 8.5 | 18.7 | 18.3 | 40.8 | 22.4 |
| | Mean token | 18.8 | 18.0 | 25.9 | 18.5 | 25.8 | 27.5 | 37.3 | 24.5 |
| | Ours | 67.7 | 56.3 | 66.5 | 53.0 | 67.4 | 75.5 | 73.5 | **65.7** |
| LLaMA-13B | Cond. Likelihood | 44.3 | 20.8 | 51.8 | 38.6 | 56.0 | 50.9 | 56.7 | 45.6 |
| | Joint Likelihood | 36.7 | 1.1 | 35.0 | 27.7 | 33.0 | 32.4 | 48.0 | 30.6 |
| | Last token | 18.2 | 24.2 | 29.0 | 16.8 | 21.9 | 10.2 | 40.8 | 23.0 |
| | Mean token | 28.8 | 25.2 | 30.2 | 20.2 | 31.5 | 35.1 | 45.0 | 30.9 |
| | Ours | 70.6 | 52.5 | 65.9 | 53.2 | 67.8 | 74.1 | 73.0 | **65.3** |
| LLaMA-33B | Cond. Likelihood | 31.4 | 21.5 | 41.5 | 35.3 | 38.8 | 38.3 | 56.3 | 37.6 |
| | Joint Likelihood | 36.2 | 4.9 | 35.6 | 27.7 | 30.3 | 32.3 | 47.8 | 30.7 |
| | Last token | 21.8 | 20.1 | 13.2 | 9.4 | 22.5 | 11.5 | 40.8 | 19.9 |
| | Mean token | 27.9 | 24.0 | 29.6 | 21.7 | 35.4 | 34.6 | 43.5 | 31.0 |
| | Ours | 71.5 | 52.5 | 70.6 | 54.6 | 69.1 | 75.2 | 73.0 | **66.6** |
| Vicuna-13B | Ours | 70.2 | 53.4 | 62.4 | 52.0 | 68.5 | 73.9 | 75.3 | 65.1 |
| StableVicuna-13B | Ours | 70.5 | 56.2 | 63.9 | 52.5 | 67.9 | 74.8 | 75.3 | 65.9 |

For experiments on WordNet hyponym/hypernym relations, we leverage the WikiText (Merity et al., 2016) corpus by sampling up to $n = 100$ contexts (i.e. paragraphs in the WikiText dataset) containing each given word.

## C.1 WORDNET HYPONYM/HYPERNYM SUBSET:

For our experiments, we use a total of 166 pairwise hyponym/hypernym relation between the following sets sampled from WordNet (Miller, 1995), enumerated in order of meaning containment:

1. {puppy, dog, canine, carnivore, predator, animal, organism}
2. {storybook, book, publication, work, product, creation, artifact}
3. {dine, eat, consume}
4. {soar, fly, travel}
5. {chuckle, laugh, express emotion}
6. {bobcat, lynx, wildcat, cat, feline, carnivore}
7. {penthouse, apartment, housing, structure}
8. {recliner, armchair, chair, seat, furniture, furnishing, instrumentality}
9. {neurosurgeon, surgeon, doctor, medical practitioner, professional, adult, person}
10. {brunch, meal, food, substance}
11. {hydrofoil, speedboat, motorboat, boat, vessel, craft, vehicle, conveyance}

12. { consult, research, investigate, analyze }

13. { symposium, conference, meeting, gathering}

14. { hacker, programmer, engineer, person}

## C.2 HYPONYM TEST

We detail the Hyponym Test for quantifying meaning containment between words in algorithm 2.

---

**Algorithm 2** Hyponym Test (Words)

---

**Require:** Model $M$, Words $u$ and $v$, number of trajectories $n$, distance $d$, Text Corpus $D_{corpus}$

$T_u = \{(s_i, t_i)\}_{i=1}^{n} \leftarrow$ Sample $n$ paragraphs $(s_i\, u\, t_i)$ containing $u$ from $D_{corpus}$

$T_v = \{(s_i, t_i)\}_{i=n+1}^{2n} \leftarrow$ Sample $n$ paragraphs $(s_i\, v\, t_i)$ containing $v$ from $D_{corpus}$

$T \leftarrow T_u \sqcup T_v$

Initialize $\overline{M}_u = \overline{M}_v = \overline{M}_u \wedge \overline{M}_v = \emptyset$

**for** $t = (a_1 \ldots a_{m_t}, b_1 \ldots b_{m'_t}) \in T_u \sqcup T_v$ **do**

$\quad \overline{M}_u[t] \leftarrow \big( \prod_{i=1}^{m_t} P_M(a_i | a_1 \ldots a_{i-1}) \cdot P_M(u | a_1 \ldots a_{m_t}) \cdot$
$$\prod_{i=1}^{m'_t} P_M(b_i | a_1 \ldots a_{m_t}\, u\, b_1 \ldots b_{i-1}) \big)^{1/(m_t + m'_t + 1)}$$

$\quad \overline{M}_v[t] \leftarrow \big( \prod_{i=1}^{m_t} P_M(a_i | a_1 \ldots a_{i-1}) \cdot P_M(v | a_1 \ldots a_{m_t}) \cdot$
$$\prod_{i=1}^{m'_t} P_M(b_i | a_1 \ldots a_{m_t}\, v\, b_1 \ldots b_{i-1}) \big)^{1/(m_t + m'_t + 1)}$$

$\quad \overline{M}_u[i] \wedge \overline{M}_v[t] \leftarrow \min(\overline{M}_u[t], \overline{M}_v[t])$

**end for**

**if** $d(\overline{M}_u, \overline{M}_u \wedge \overline{M}_v) < d(\overline{M}_v, \overline{M}_u \wedge \overline{M}_v)$ **then**

$\quad$ **return** $u$ hyponym of $v$

**else**

$\quad$ **return** $v$ hyponym of $u$

**end if**

---

# D ADDITIONAL VISUALIZATIONS

## D.1 PERFORMANCE SCALES WITH MODEL SIZE

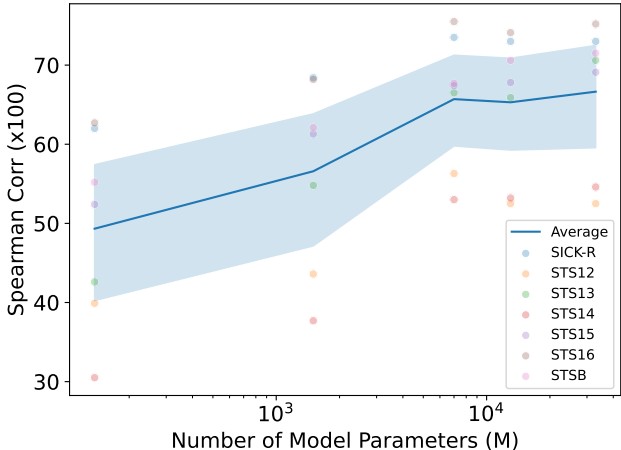

Figure 4: Plot of performance on semantic textual tasks vs number of model parameters, as measured using GPT-2, GPT-2-XL, Falcon-7B, LLaMA-13B, and LLaMA-33B.

We show in Figure 4 that the alignment of our method on semantic textual similarity with human annotators scales with model size.

In Figure 5, we show further visualizations of the hyponym/hypernym hierarchies established by our method on WordNet using Falcon-7B.

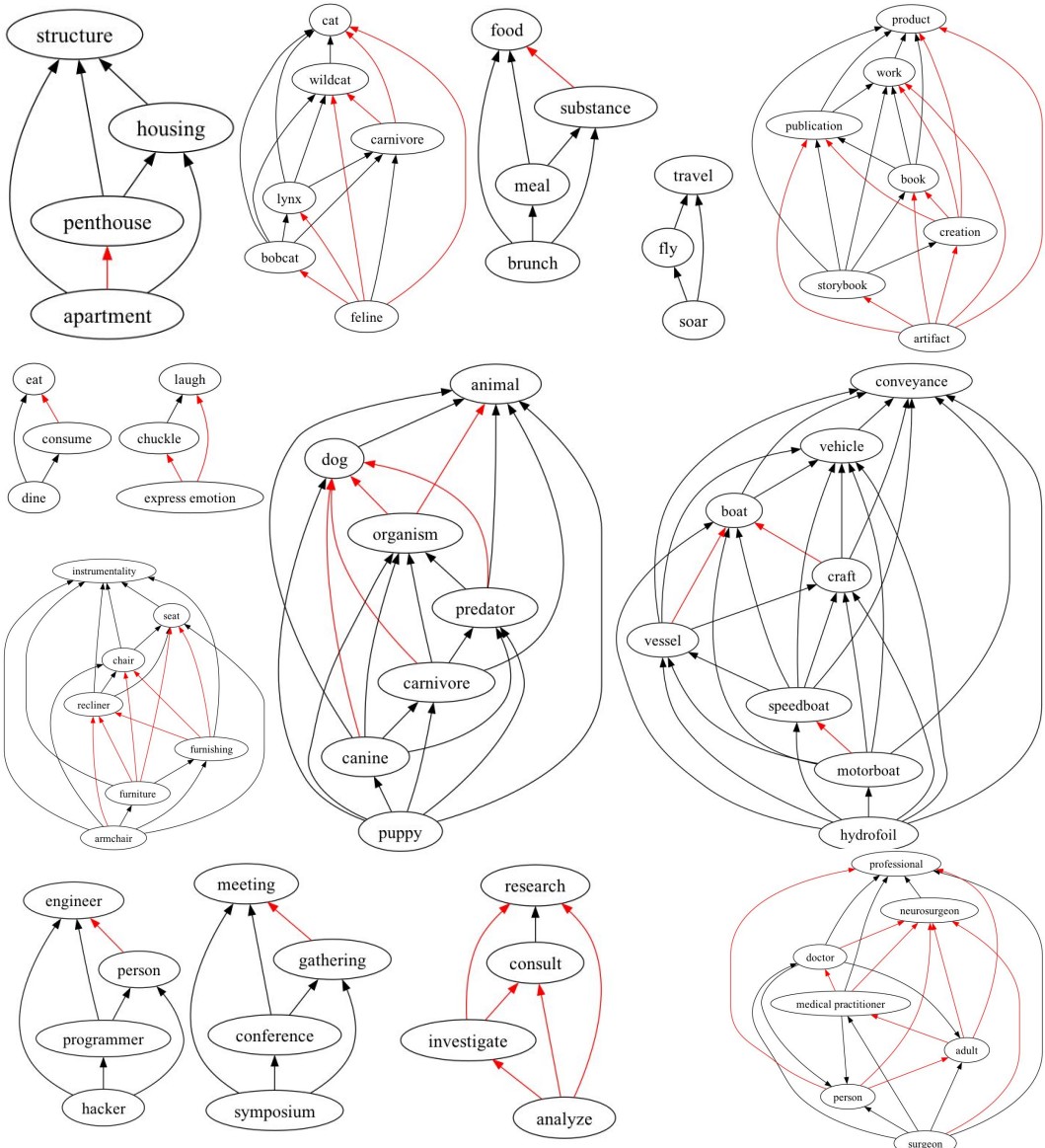

Figure 5: WordNet Hyponym/Hypernym Relation predictions using Falcon-7B

# E    PROMPTING FOR DOWNSTREAM TASKS

Our definition of meaning in the context of large language is prompt-free, and hence not subject to the drawbacks and variabilities that arise from prompt-engineering. However, prompts can naturally be used to improve performances on downstream tasks by conditioning the trajectories obtained from input strings.

Table 7: We implemented Jiang et al. (2023) for LLaMA-13B, and show that prompt-based methods are brittle and generalize poorly to other architectures apart from that which they were tuned on. Nevertheless, we show that prompting, while detracting from retaining our "pure" notion of meaning, can be used to further improve our results on downstream semantic textual similarity tasks. We also compare against other prompt-based methods here. For Ours-Prompt-1, we simply prepend "The meaning of this sentence is: " to each input string. For Ours-Prompt-2, we append "This sentence implies " to the end of each input string to condition the set of trajectories towards logical implications, achieving superior results across zero-shot, prompt-based methods. † indicates results taken from Jiang et al. (2023).

| Method | STS-B | STS12 | STS13 | STS14 | STS15 | STS16 | SICK-R | Avg |
|---|---|---|---|---|---|---|---|---|
| Contrastive-Trained Models | | | | | | | | |
| PromptBERT (Jiang et al., 2022) | 81.6 | 71.6 | 84.6 | 77.0 | 84.5 | 80.6 | 69.9 | 78.5 |
| PromptRoBERT (Jiang et al., 2022) | 81.9 | 73.9 | 84.7 | 77.3 | 85.0 | 81.7 | 69.5 | 79.2 |
| Autoregressive Models | | | | | | | | |
| PromptEOL (OPT-1.3B)[†] | 73.2 | 64.6 | 79.1 | 68.5 | 78.9 | 78.6 | 69.4 | 73.2 |
| PromptEOL (OPT-13B)[†] | 70.7 | 60.2 | 81.4 | 67.0 | 75.5 | 79.6 | 66.0 | 71.9 |
| PromptEOL (OPT-66B)[†] | 71.7 | 55.7 | 74.6 | 64.9 | 72.3 | 75.2 | 67.4 | 68.8 |
| PromptEOL (LLaMA-13B) | 63.4 | 52.3 | 75.3 | 64.0 | 70.5 | 73.2 | 60.5 | 65.6 |
| Ours (LLaMA-13B) | 70.6 | 52.5 | 65.9 | 53.2 | 67.8 | 74.1 | 73.0 | 65.3 |
| Ours-Prompt-1 (LLaMA-13B) | 72.2 | 61.6 | 68.4 | 66.9 | 72.7 | 75.6 | 76.3 | 70.5 |
| Ours-Prompt-2 (LLaMA-13B) | 81.5 | 67.9 | 79.9 | 75.3 | 82.9 | 82.3 | 74.6 | 77.8 |

### E.1 PROMPTING FOR SEMANTIC TEXTUAL SIMILARITY

By implementing an existing prompt-based method (Jiang et al., 2023) on LLaMA-13B, we show in Table 7 that prompt-based methods are brittle and model-specific, hence require careful tuning for each specific architecture. In contrast, our original method is prompt-free and robust against such variances arising from prompt-engineering.

Nevertheless, we present some preliminary investigations for augmenting our method with prompts in Table 7 for the STS task. We also compare against existing zero-shot prompt-based methods. We show that prompting can also significantly improve performance over the prompt-free approach for the STS task, by appending "The meaning of this sentence is: " to each input string when generating trajectories. We note that we did not carefully search over prompts, and simply tried the first ones (above) that came to mind. It is likely that there exist others which work better.

### E.2 ALIGNMENT PROMPTS FOR VISION-LANGUAGE MODELS

In the main paper, we presented prompt-free approaches for extracting similarity scores from multimodal inputs. However, we note that by our definitions, LLaVA (Liu et al., 2023) does not technically attribute the same meaning to images and captions. We visualize this in Figure 6, where we show that image and caption inputs are continued very differently by the model. For instance, given an image, LLaVA generally attempts to generate a caption. On the other hand, when given a caption, LLaVA simply continues it, often in an unpredictable manner. In spite of this misalignment, there exists sufficient overlap in likelihood scores to outperform all baselines as observed in Table 2 of the main paper.

We demonstrate that a prompt can optionally be used to align the meaning representations for vision and text modalities for the purposes of semantic comparison. We achieve this by conditioning the caption continuations to match the continuations of images. In particular, we append "*This is a caption for an image. Describe this image. This image shows*" to the text inputs, and "*Describe this image. This image shows*" to image inputs. Figure 6 (Right) shows that this successfully aligns the trajectories of both modalities. Indeed Table 2 of the main paper shows that this improves over the prompt-free version of our method on the CxC-SITS task by 18.4%, and outperforms the CLIP paragon by 13.3%.

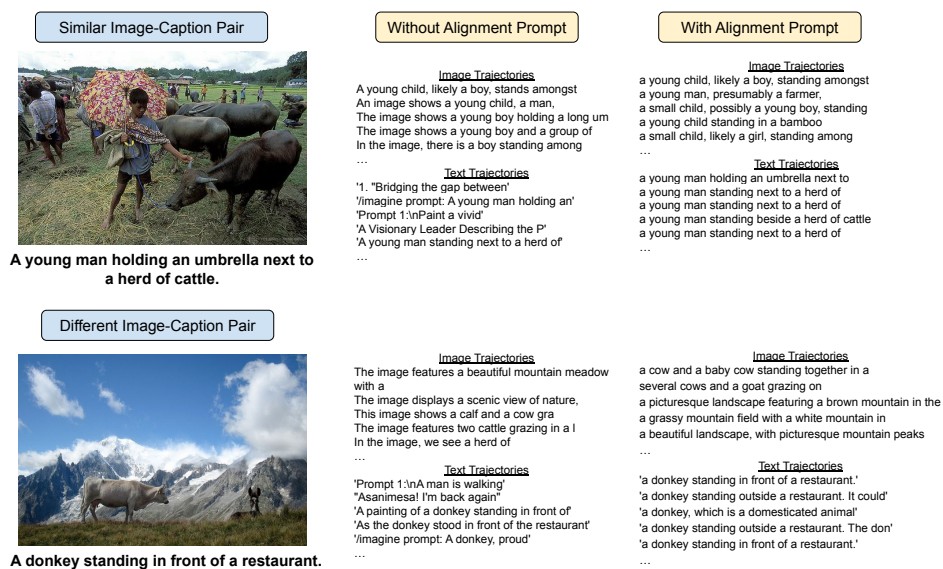

Figure 6: A prompt can be used to align the meaning representations (i.e., distribution over trajectories) for vision and text modalities to measure image-caption similarities. We obtain trajectories on the right by appending "*Describe this image. This image shows*" to image inputs, and appending "*This is a caption for an image. Describe this image. This image shows*" to caption text inputs.

## F  MEANING CONTAINMENTS

In this section, we discuss how the partial ordering defined on our meaning representations is related to entailment ($\Rightarrow$) between statements and to hyponym/hypernym relations between words.

### F.1  ENTAILMENT TEST

In the main body of the paper, we claimed that if $u \Rightarrow v$ then $M_v \leq M_u$ is more natural than $M_u \leq M_v$. Our empirical experiments indeed show that our Entailment Test succeeds significantly more often than chance. Intuitively, this means that if $u \Rightarrow v$ then more continuations for $v$ are feasible continuations for $u$, instead of the other way around.

To understand why this is the case, we consider the sets $C_u, C_v$ of *consequents* of $u$ and $v$, that is, the set of sentences $t$ such that $u \Rightarrow t$ or $v \Rightarrow t$ respectively. If $u \Rightarrow v$, then by transitivity of entailment we have $C_v \subset C_u$ (since $v \Rightarrow t$ implies $u \Rightarrow t$). Thinking $M_u$ and $M_v$ as sets, then if $M_u = C_u$ and $M_v = C_v$ were true (i.e., if the set of feasible continuations coincided with the set consequents), then this would justify our claim. In practice, continuations and consequents do not coincide, especially because some continuations are not consequents. However, it is generally true that consequents are valid continuations—so approximately $C_u \subset M_u, C_v \subset M_v$—and overall consequents seem sufficiently frequent as continuations to dictate the containment relation among general continuations.

To make this argument more concrete, consider the sentences $u =$"*Cody, the neighbor's dog, is barking.*" and $v =$"*A dog is barking.*", so that $u \Rightarrow v$. If $t$ is a continuation of $v$, then $t$ could in general be

- a consequent of $v$ ($v \Rightarrow t$) for example $t =$"*Therefore, I can't sleep.*"

- an antecedent of $v$ ($t \Rightarrow v$) for example $t =$"*Indeed, there is a cat.*".

- non-comparable with $v$ (neither $v \Rightarrow t$ nor $t \Rightarrow v$ hold) for example, "*The dog is brown.*" or "*The dog's name is Spot*".

From these examples, we see that: 1) consequent continuations for $v$ are also valid continuations for $u$; 2) antecedent continuations are somewhat unnatural and likely uncommon; 3) non-comparable

continuations of $v$ may or may not be valid continuations of $u$. Overall, if we think of consequent continuations as the default, then we expect the containment direction $M_v \leq M_u$ to hold more than $M_u \leq M_v$

## F.2 HYPONYM TEST

In the paper, we claimed that if $v$ is a hyponym of $u$ then $\overline{M}_v \leq \overline{M}_u$ is a more natural relation than $\overline{M}_v \leq \overline{M}_u$. Our experiments indeed suggest that $\overline{M}_v \leq \overline{M}_u$ occurs significantly more often than chance. Intuitively, this means that if $v$ is a hyponym of $u$ then $v$ can be substituted with $u$ more often than the other way around.

To investigate why this is the case, we distinguish between two types of usages of a common noun $v$ (e.g., "dog"):

- Definite reference: when $v$ refers to a specific instance or set of instances of the noun, for example *"The dog is barking."*
- Generic reference: when $v$ refers to *all* instances of the noun, for example *"Any dog is an animal"*.

Now, if $s$ is a sentence that uses a $v$ with definite reference, then it is possible to replace $v$ with a hypernym in $s$ (*"The dog is barking."*→*"The animal is barking."*). In contrast, if $s$ uses $v$ with generic reference, then we can replace $v$ with a hyponym (*"Any dog is an animal."*→*"Any German Shepherd is an animal."*). Thus, whether hyponym relations correspond to $\overline{M}_v \leq \overline{M}_u$ or $\overline{M}_u \leq \overline{M}_v$ depends on which type of reference is more common. Our empirical results suggest that definite reference is more common—as one might probably expect, particularly for singular nouns.

We note that in practice, $M_u \leq M_v$ rarely happens. As an alternative, we can express this relation as $d(M_u \wedge M_v, M_u) = 0$, where $\wedge$ represents the meet operation given by $M_u \wedge M_v := \min(M_u, M_v)$. In other words, $M_u \leq M_v$ if $M_u$ is contained within their intersection. This alternative formulation offers a crucial advantage, as it provides a soft measure of containment that quantifies the strength of this relation.

## G LANGUAGES AND MEANINGS

In this section, we present a more theoretical discussion that motivates our notion of meaning representation. We also introduce some definitions and perspectives on LLMs that were not required for describing the methods proposed in the paper but that may be of independent interest. We recall that in the main body of the paper, we identified a language model with a map $M : \mathcal{A}^* \times \mathcal{A}^* \to [0, 1]$. Here we take a step back and start from a more primitive notion of autoregressive token generator.

**Definition 1.** *An* autoregressive token generator *is a map* $G : \mathcal{A}^* \times \mathcal{A} \to [0, 1]$ *associating any string $s$ with a score $G(a|s)$ for identifying the next token $a$ in $\mathcal{A}$. Given any $u \in \mathcal{A}^*$, we use $G_u$ to denote the* prompted token generator, *that is, a token generator defined by $G_u(a|s) := G(a|us)$. We write $\mathcal{G}(\mathcal{A})$ for the set of all autoregressive token generators with tokens from $\mathcal{A}$.*

Starting with an initial prompt $u_0$, a greedy text generation process using the generator $G$ returns a sequence of strings $u_{i+1} = u_i a$, or a *trajectory*, where $a$ is a token recovered from $G(u_i)$ according to some sampling scheme.

Given a candidate trajectory $u = a_1 \ldots a_n$, the token generator provides a sequence of token-level scores $G(a_1 \ldots a_{i-1})(a_i)$ in $[0, 1]$. These scores can be aggregated, for example by simply taking their product. In practice, it is more common to normalize by sequence length. Thus, we consider the sequence level score as given by

$$\prod_{i=1}^{n} G(a_i|a_1 \ldots a_{i-1})^{1/n}. \tag{4}$$

This choice allows us to use a generator evaluate candidate trajectories, obtaining a map $\mathcal{A}^* \to [0, 1]$. We think of such a map as a (soft) "predicate" characterizing strings that the model considers "feasible."

**Definition 2.** *A* linguistic predicate *is a map* $L : \mathcal{A}^* \to [0,1]$. *We write* $\mathcal{L}(A)$ *for the set of all linguistic predicates with tokens from* $\mathcal{A}$.

Any autoregressive language generator $G$ can thus be uniquely associated with a linguistic predicate $L(G) \in \mathcal{L}(\mathcal{A})$ using eq. (4). Conversely, any linguistic predicate $L$ determines an associated token generator $G(L) \in \mathcal{G}(\mathcal{L})$ by setting

$$G(L)(a|u) = \min\left(\frac{L(u\,a)^{|u|+1}}{L(u)^{|u|}}, 1\right),$$

where this minimum is taken to be 1 whenever the denominator of the left term is zero. Thus, we obtain two "dual" maps $L : \mathcal{G}(\mathcal{A}) \to \mathcal{L}(\mathcal{A})$ and $G : \mathcal{L}(\mathcal{A}) \to \mathcal{G}(\mathcal{A})$. These maps are not full inverses but satisfy $G \circ L = Id_{\mathcal{G}}$.

A generator $G$ also uniquely determines a language model $M_G : \mathcal{A}^* \times \mathcal{A}^* \to [0,1]$, as defined in the main body of the paper, by simply setting $M_B(s|u) := L(G_u)(s)$. Using these definitions, the meaning representation of the prompt $u$ for the token generator $G$ (or the model $M_G$) is the linguistic predicate $L(G_u)$ associated with the prompted generator.

As mentioned in the paper, these ideas are closely connected to the theory of automata and formal languages. To see this, consider a "crisp" token generator $G$ whose values are always in $\{0,1\}$. In this setting, we say that a string $u = a_1 \ldots a_n$ (or trajectory) is "feasible" if $G(a_i|a_1 \ldots a_{i-1}) = 1$ for all $i$. The associated predicate $L(G)$ also takes values in $\{0,1\}$ and can be seen as the formal language consisting of all feasible strings (we identify $\{0,1\}$-valued functions with subsets of the domain).[1] We now remark that, *if a string $u$ is acceptable* for $G$, then

$$L(G_u) = u^{-1}L(G),$$

where $u^{-1}L := \{v \in \mathcal{A}^* : uv \in L\}$ is the "left quotient" of $L$ by $u$ (sometimes also known as the *Brzozowski derivative*). The set $u^{-1}L$ is a class of the equivalence relation on $\mathcal{A}^*$ given by

$$u_1 \sim_L u_2 \Leftrightarrow u_1 s \in L \text{ iff } u_2 s \in L.$$

Thus, $u_1 \sim_L u_2$ if $u_1$ and $u_2$ have no "distinguishing continuations." This equivalence relation features in the construction of the *minimal automaton* that accepts a given language Pin (2022). More precisely, for any language $L$, a minimal automaton for $L$ has states identified with left quotients $\{u^{-1}L : s \in \mathcal{A}^*\}$, accepting states corresponding $F = \{s^{-1}L : s \in L\}$ (classes of strings in $L$), and actions for each token $a \in \mathcal{A}$ described by

$$(s^{-1}L) \cdot a = (sa)^{-1}L.$$

Thus, the meaning representations $L(G_u) = u^{-1}L(G)$ for acceptable strings $u$ correspond exactly to the states of a minimal automaton accepting the language $L(G)$. We also remark that a different equivalence relation $L$, sometimes known as the *syntactic congruence*, is given by

$$u_1 \approx_L u_2 \Leftrightarrow su_1 t \in L \text{ iff } su_2 t \in L,$$

and has the property that $u_1, u_2 \in \mathcal{A}^*$ induce the same action on states of the minimal automaton if and only if $u_1 \approx_L u_2$; in other words, the monoid of transformations on states is given by $\mathcal{A}^*/\sim_{synt}$ (Pin, 2022). The equivalence classes for this relation correspond to the meaning representation for substrings that we consider in the paper, a refinement of the meaning representation for prefixes.

**Remark 3.** *If a string $u$ is not feasible for a crisp generator $G$, then we have $u^{-1}L(G) = \varnothing$, since the product in eq. (4) is zero when at least one token is not acceptable. On the other hand, according to Definition 1, the language $L(G_u)$ depends only on tokens following $u$, and thus may a priori be* arbitrary *and* unrelated *to the language $L(G)$. Practically, this means that* infeasible prompts may lead to completely unpredictable continuations. *This intuition may also be useful for general (non-crisp) language models, by thinking of infeasible prompts as strings with very low likelihood for the model.*

---

[1]According to our definitions, the language $L(G)$ associated with a generator is always a prefix-closed set. If the vocabulary $\mathcal{A}$ has a "end of sentence" [EOS] token, one could also consider the language of all strings that are feasible and also "complete," i.e., such that $G([EOS]|u) = 1$.

We conclude by revisiting these ideas in the actual $[0,1]$-valued setting considered in the paper. To do so, we take a "coalgebraic" perspective on automata, as described in (Jacobs, 2012). We view a deterministic automaton with $[0,1]$-valued outputs and action set $\mathcal{A}$ as a triple $(S, \delta, \lambda)$ where $S$ is a set and $\delta, \lambda$ are maps

$$\delta : S \times \mathcal{A} \to S, \qquad \lambda : S \to [0,1].$$

Here $\delta$ describes state transitions and $\lambda$ describes (soft) acceptance of states (note that we do not model the initial state; for this reason we sometimes also use the term "process" instead of automaton). An autoregressive token generator $G$ can be seen as an automaton in which $S = \mathcal{A}^*$, $\delta$ is string concatenation, and $\lambda$ is the predicate $L(G)$.

Given two automata $(S, \delta, \lambda)$ and $(T, \delta', \lambda')$, a morphism between the two is defined by a map between states $f : S \to T$ such that

$$f(\delta(s, a)) = \delta'(f(s), a) \quad \text{and} \quad \lambda(s) = \lambda'(f(s)), \quad \forall s \in S, a \in \mathcal{A}.$$

We write such a morphism as $f : (S, \delta, \lambda) \to (T, \delta', \lambda')$.

Intuitively, a "semantic interpretation" of an automaton $(S, \delta, v)$ is given by a morphism $m : (S, \delta, \lambda) \to (U, \gamma, \mu)$, where $U$ is some sort of "meaning space" and $\gamma, \mu$ correspond transitions and observations of $S$ within $U$.[1] Moreover, a desirable property would be that the meaning process $(U, \gamma, \mu)$ is also *universal*: this would mean that *any* automaton $(S, \delta, \lambda)$ can be interpreted in $(U, \gamma, \mu)$ and *in a unique way*. The following standard result shows such a process actually exists and its states correspond to predicates.

**Proposition 4** (Proposition 2.3.5 Jacobs (2012)). *Let $(U, \gamma, \mu)$ be the process given by $U = [0,1]^{\mathcal{A}^*}$ and for any $L \in U$:*

$$\gamma(L)(a) = L_a, \text{ where } L_a(s) := L(as), \qquad \mu(L) = L(\epsilon),$$

*where $\epsilon$ is the empty string. Then, for any automaton $(S, \delta, \lambda)$, there exists a unique morphism $m : (S, \delta, \lambda) \to (U, \gamma, \mu)$. Moreover, $(U, \gamma, \mu)$ is determined (up to isomorphism) by this property.*

The unique map $m : (S, \delta, \lambda) \to (U, \gamma, \mu)$ from this Proposition can be described as

$$S \to U = [0,1]^{\mathcal{A}^*}, \qquad s \mapsto (v \mapsto \lambda(\delta^*(s, v))), \qquad s \in S, v \in \mathcal{A}^*,$$

where $\delta^* : S \times \mathcal{A}^* \to S$ is the interated transition function. In particular, if we consider the process $(\mathcal{A}^*, \cdot, L(G))$ associated with a token generator $G$, this unique map is

$$u \mapsto (v \mapsto L(G)(uv)), \qquad u, v \in \mathcal{A}^*.$$

The association on the right is analogous to the left-quotient set $u^{-1}L(G)$ considered for crisp models and motivates the semantic representation considered in this work.

---

[1]This kind of interpretation assumes that the set $\mathcal{A}$ also acts on meanings. When modeling natural language, it is probably more natural to think of $\mathcal{A}$ as some collection of "meaningful sentences" (as opposed to tokens) so that the state space $S = \mathcal{A}^*$ of a language generator is the set of concatenations of such sentences. This would not significantly change our discussion nor the construction of our representations.

