# OpenReview forum: "Meaning Representations from Trajectories in Autoregressive Models"
_ICLR.cc/2024/Conference — ICLR 2024 poster_

### Official Review · Reviewer_xcZs · 2023-10-30

**Soundness:** 4 excellent
**Presentation:** 4 excellent
**Contribution:** 3 good
**Rating:** 8
**Confidence:** 4

**Summary:**

This paper proposes a "meaning representation" of a text based on inverse perplexity for the continuation sequence of the text.
The authors define a semantic distance between two prompts using their meaning representations.
This is useful for capturing the similarity between texts and for testing hyponym relationships.
Additionally, it can be applied to multimodal autoregressive models.

**Strengths:**

This paper is well-written, featuring concrete formulations and structured experiments.
The authors introduce a novel and powerful semantic distance measurement that captures sentence similarity and hyponym relationships. Furthermore, they demonstrate that this distance can be computed more efficiently than initially concerned, alleviating worries about sampling a lot of trajectories.

**Weaknesses:**

1) The authors define a 'syntactic meaning representation' as the function $M_s$. I'm unclear as to why it is named 'syntactic meaning representation' (I'm also unsure why 'syntactic' is included). It is simply the conditional probability of the prompt. I don't find it to be a useful 'representation' for training or other applications like vector-space representation. In fact, they merely use the divergence between the conditional probabilities to measure text similarity. If this approach hasn't been taken by others, then this divergence could be considered novel. Thus, I cautiously suggest they use 'inverse perplexity mapping' rather than 'meaning representation'. The paper's title could then be 'A New Measurement for Sentence Similarity via Sampling Trajectories in Autoregressive Models'. Furthermore, 'semantic representations' appear suddenly following Algorithm 1. I recommend they use 'inverse perplexity mapping for substrings' instead of 'semantic representations'.

2) I'm unsure what is meant by 'meaning containment'. Why use 'containment', which typically refers to 'the action of keeping something harmful under control or within limits'? Please provide a definition or explanation, along with some references.

3) For the final version, sharing the code on GitHub would be beneficial for readers.

**Questions:**

1) The similarity distance depends on the performance of the autoregressive model. If the model is fine-tuned, e.g., for a chatbot, then how would the similarity change?

2) In the first sentence of the second paragraph under **Meaning representation for prompts**, I think $M_u$ should be changed to $M_s$ and $t\in \mathcal{A}^1$ should be changed to $t\in \mathcal{A}^*$.

3) In the second sentence of the paragraph **Containments of semantic representations**, the definition of 'partial order' should be revised. Since $\sum_{\mathrm{len}(t) = m} M_u(t)^{m} = \sum_{\mathrm{len}(t) = m} M_v (t)^{m} = 1$, we cannot have $M_u(t) < M_v(t)$ for all $t\in \mathcal{A}^*$. Please update the definition of 'partial order.'

4) What is the temperature $\lambda$ in the experiment? Is it the temperature parameter used during the inference process in the autoregressive model? Please clarify this.

5) I believe we should use the training set for the model to compute $\overline{M}_u$. Does the text corpus WikiText reflect the distribution of the training set?

---

> ### Author Response · Authors · 2023-11-15
> **Response to Reviewer xcZs [Part 1 of 2]**
>
> We thank the reviewer for their thoughtful feedback and suggestions. We address each concern in detail below.
>
> >**The authors define a 'syntactic meaning representation' as the function $M_s$. I'm unclear as to why it is named 'syntactic meaning representation' (I'm also unsure why 'syntactic' is included).**
>
> We included “syntactic” to stress that our definition of meaning representation is completely determined by syntactic/formal properties of natural language (_i.e._, what strings can extend other strings), as opposed to being defined in terms of external referents. It is also consistent with standard definitions in the theory of formal languages (https://en.wikipedia.org/wiki/Syntactic_monoid) which capture the spirit of our perspective.
>
> >**It is simply the conditional probability of the prompt**
>
> Since we define this more generally to be the score distribution over all possible continuations of the given prompt, the score attribution mechanism is not required to be the conditional probability distribution over these trajectories. In our experiments, we use inverse perplexity, which is also not an actual probability (since does not sum to 1).
>
> The reason we choose this more general definition is to allow composition operations in this space of meanings that might not be obtainable from any individual prompt. One example, which we use when measuring asymmetric relations, is $M_u \land M_v$. The resulting distribution will in general not be the “conditional probability” of any particular prompt, yet resides in the same geometric space which allows computation of distances such as $d(M_u \land M_v, M_v)$.
>
> >**I don't find it to be a useful 'representation' for training or other applications like vector-space representation.**
>
> Our representations attain strong results on downstream tasks. In contrast, previous methods for extracting representations from autoregressive models are less effective, as shown in our experiments.
>
> In general, there is no formal difference between the expressive power of our representations and vector-space representations obtained from traditional embedding models like CLIP: any task that can be performed using CLIP embeddings can also be accomplished with ours (albeit with greater computational cost). Indeed, any method that measures semantic similarity between arbitrary pairs of input texts actually provides an implicit semantic representation (which could potentially be made explicit via multidimensional scaling methods). It can also be used for classification tasks directly using kernel methods. Furthermore, as argued above, our representation is also naturally interpretable, since coordinates correspond to trajectories.
>
> Lastly, our experiments on the SICK-R dataset in the revised draft also show that our representations handle compositional knowledge even better than vector-space representations from contrastive trained models.
>
> >**In fact, they merely use the divergence between the conditional probabilities to measure text similarity. If this approach hasn't been taken by others, then this divergence could be considered novel.**
>
> Divergence between conditional probabilities of next-token predictions is a natural baseline that is commonly used in existing fine-tuning or prompt-based methods. In contrast, we compute the distance between score distributions over trajectories as a similarity measure, which has never been adopted by any prior work to the best of our knowledge.
>
> >**The paper's title could then be 'A New Measurement for Sentence Similarity via Sampling Trajectories in Autoregressive Models'.**
>
> We appreciate the reviewer’s suggestion, but we believe our current title is a more faithful representation of our work for the abovementioned, and the following reasons:
>
> 1. We propose a working definition of meaning representation for autoregressive models. Quantifying semantic similarity is just one out of the applications enabled by our proposed definition. We demonstrate its versatility by applying it to tasks involving asymmetric relations, such as quantifying entailment between sentences and hypernym/hyponym relations between words.
>
> 2. The applicability of our method extends beyond sentence comparison, as it was successfully used with a broader range of autoregressive models, including vision-language models.
>
> 3. We do not compare sampled trajectories, but rather compare distributions over trajectories. This distinction is crucial, particularly when comparing meaning representations of words, which we define as the distribution of scores over contexts in which the words appear. Hence even though such contexts cannot be sampled directly, our proposed meaning representation remains well-defined and applicable, allowing for alternative methods of obtaining trajectories such as context retrieval from a corpus.

---

> > ### Author Response · Authors · 2023-11-15
> > **Response to Reviewer xcZs [Part 2 of 2]**
> >
> > >**Furthermore, 'semantic representations' appear suddenly following Algorithm 1. I recommend they use 'inverse perplexity mapping for substrings' instead of 'semantic representations'.**
> >
> > Our choice of terminology is meant to be consistent with that used for encoder-based models such as CLIP, since the representations we obtain can be used in a similar manner.
> >
> > >**I'm unsure what is meant by 'meaning containment'. Why use 'containment', which typically refers to 'the action of keeping something harmful under control or within limits'? Please provide a definition or explanation, along with some references.**
> >
> > We use the term “containment” in the set-theoretic sense (A is contained in B if $A \subseteq B$). We refer the reviewer to Section F of the Appendix, which is dedicated to discussing and motivating this terminology.
> >
> > >**For the final version, sharing the code on GitHub would be beneficial for readers.**
> >
> > We agree and indeed plan to release our code alongside the final version.
> >
> > >**The similarity distance depends on the performance of the autoregressive model. If the model is fine-tuned, e.g., for a chatbot, then how would the similarity change?**
> >
> > Great question! We believe that fine-tuning primarily focuses on syntactic alignment with human instructions rather than semantic understanding. For instance, a model trained to output "2" more frequently than "3-1" in response to "1+1=" might not necessarily indicate an improved understanding of the equation but rather an enhanced ability to format its output to match human expectations for a chatbot. While fine-tuning does alter meaning representations, it does not guarantee better alignment with human annotations on semantic similarity tasks. This is demonstrated in Appendix B, Table 6, where we compare LLaMA-13B with its instruction-tuned and RLHF variants – Vicuna and StableVicuna, respectively – revealing that all three models exhibit comparable performance on semantic similarity tasks.
> >
> > >**In the first sentence of the second paragraph under Meaning representation for prompts, I think $M_u$ should be changed to $M_s$ and $t \in A^1$ should be changed to $t \in A^\ast$**
> >
> > $M_u → M_s$: Thank you for catching this, we have updated our draft.
> >
> > $t \in A^1$: This is intentional, since we were referring to a special case of Equation (1) where the maximum length of the sampled trajectories is restricted to 1 token.
> >
> > >**In the second sentence of the paragraph Containments of semantic representations, the definition of 'partial order' should be revised. Since $\sum_{\mathrm{len}(t) = m} M_u(t)^{m} = \sum_{\mathrm{len}(t) = m} M_v (t)^{m} = 1$, we cannot have $M_u(t) < M_v(t)$ for all $t \in A^\ast$. Please update the definition of 'partial order.'**
> >
> > Thank you for pointing this out, we clarified our definition in the revised draft. Indeed, this is precisely the reason for leveraging the soft notion of containment $d(M_u \land M_v, M_u)$ to quantify this relation. Note that the composition $M_u \land M_v$ also lives in the same space of meanings, and the partial ordering $M_u \land M_v \leq M_u$ and  $M_u \land M_v \leq M_v$ always holds by definition.
> >
> > >**What is the temperature $\lambda$ in the experiment? Is it the temperature parameter used during the inference process in the autoregressive model? Please clarify this.**
> >
> > Yes, we have updated our draft to clarify this, thank you.
> >
> > >**I believe we should use the training set for the model to compute $\overline{M}_u$. Does the text corpus WikiText reflect the distribution of the training set?**
> >
> > Due to limitations in accessing the full data distributions used to train many of the large language models employed, we utilized WikiText as our corpus. We empirically found that it is sufficiently large, and contains enough contexts to evaluate the meaning representation of the words we tested. Nevertheless, we concur with the reviewer's suggestion that larger and more relevant corpora would further improve evaluation.

---

> > > ### Comment · Reviewer_xcZs · 2023-11-22
> > > **Reviewer Response**
> > >
> > > Thank you for your clarification! I'll maintain my rating.

---

### Official Review · Reviewer_7D5R · 2023-10-31

**Soundness:** 3 good
**Presentation:** 3 good
**Contribution:** 3 good
**Rating:** 6
**Confidence:** 3

**Summary:**

This work proposes an alternative sentence meaning representation (specifically for sentence similarity) for zero-shot use with auto-regressive LMs.   The method is to sample N possible following texts for two sentences A and B, and score each of those 2N following text for its probability of following A and the probability of following B, and then compare difference between the probabilities. They explore the ability to modify the notion of similarity by augmenting those sentences with prompts, and evaluate on zero-shot Semeval STS tasks, a task of comparing captions to CLIP outputs, and (using a modified approach to focus on single-word characterization) a lexical semantics task (WordNet hypernym modeling).

**Strengths:**

- The work seems to outperform other methods for autoregressive representation of sentences on the sentence similarity tasks studied, indicating its potential for continued relevance and utility.
- The ability to modify similarity using prompting is very clever.
- On a theoretical level, thinking about sentence meaning in terms of this theoretical notion of a trajectory of meanings is a great framing.

**Weaknesses:**

- It has a relatively rigid and narrow use case, since this method can only be used for pairwise comparison and since it's not obvious how to fine-tune it.
- The work frames it as producing "interpretable" vectors, but the work was somewhat lacking in an actual exploration of that interpretability.
- I liked the idea of the entailment and hypernymy work, but it felt a bit convoluted: the way they approached both tasks seems to have lead to them comparing to weak baselines, despite NLI and wordnet link detection being well-explored areas.

**Questions:**

- As mentioned above: do the authors feel that this would be viable to fine-tune, or is it limited entirely to zero-shot STS settings (plus entailment/hypernymy)?
- Wouldn't the general hypernymy assumptions often be violated under negation?

---

> ### Author Response · Authors · 2023-11-15
> **Response to Reviewer 7D5R [Part 1 of 2]**
>
> We thank the reviewer for their valuable comments and suggestions. We respond to each comment below.
>
> >**It has a relatively rigid and narrow use case, since this method can only be used for pairwise comparison**
>
> Our representations can actually be used in much the same way as traditional embedding models such as CLIP or BERT (albeit with a greater computational cost), for example for tasks like retrieval or classification. Moreover, our representation has the additional benefit of interpretability, since unlike most embedding models, the “coordinates” used have meaningful interpretations as text continuations (trajectories).
>
> >**it's not obvious how to fine-tune it.**
>
> We note that our measure of similarity can be directly used with existing contrastive learning-based fine-tuning methods. Typical contrastive learning methods are limited to aligning next/mean-token predictions, which we show in our experiments are actually an inadequate representation of meaning.
>
> Based on this, we believe that contrastive fine-tuning with our measure of pairwise similarity could be used to better align the meaning spaces of autoregressive models with humans. Indeed, while contrastive fine-tuning methods typically sacrifice the language modeling capabilities of the LLM by converting it into an embedding model, we hypothesize that fine-tuning using our method can preserve generation abilities given appropriate regularization.
>
> While fine-tuning models for better alignment with humans are beyond the scope of our paper, we appreciate the reviewer’s suggestion and agree that it represents a valuable avenue for future research.
>
> >**The work frames it as producing "interpretable" vectors, but the work was somewhat lacking in an actual exploration of that interpretability.**
>
> Unlike traditional vector-based embeddings, where the interpretation of individual vector components is elusive, each component of our method's representations corresponds to a specific text continuation, or trajectory. This is illustrated in Figure 1 of our paper, in which each bar corresponds to the score attributed to a distinct trajectory. This provides a way to interpret the difference in meaning between sentences based on the trajectories that differentiate them. For example, in Figure 1, we observe that the sentences "A man is playing a flute" and "A dog is barking at a fly" differ because "The dog's owner is telling him to stop" is a relevant continuation for the latter but not for the former.
>
> >**I liked the idea of the entailment and hypernymy work, but it felt a bit convoluted**
>
> Our proposed evaluation methods serve as a proxy to assess the effectiveness of our entailment and hypernym/hyponym definitions. The original definitions are more general, providing a quantitative measure of the strength of these relationships between any two inputs. However, the absence of human-labeled datasets quantifying the degree of entailment between sentences or the hyponym relation between words necessitates an alternative approach, which our evaluation methods aim to address.
>
> >**the way they approached both tasks seems to have lead to them comparing to weak baselines, despite NLI and wordnet link detection being well-explored areas.**
>
> Despite the well-established nature of these tasks, they are rarely evaluated on pre-trained autoregressive models in a zero-shot setting due to the poor performance of existing methods, as rightly pointed out by the reviewer. Our findings corroborate this observation, demonstrating that existing baselines fail even for binary prediction of these relations under such settings. Our method effectively tackles this challenge.
>
> >**As mentioned above: do the authors feel that this would be viable to fine-tune, or is it limited entirely to zero-shot STS settings (plus entailment/hypernymy)?**
>
> Fine-tuning is definitely viable for the goal of aligning the meaning space of models with that of humans, and an idea for doing so using our method was described above. We would love to see future work exploring this exciting direction.
>
> It is, however, beyond the scope of this paper, since fine-tuning disrupts the meaning representation attributed by the original model, which was the subject of investigation of our work.
>
> We also present additional experiments on SICK-R in the revised draft, showing that even in the zero-shot setting, our representations effectively handle compositional knowledge, achieving even better performance than contrastive trained models.

---

> > ### Author Response · Authors · 2023-11-15
> > **Response to Reviewer 7D5R [Part 2 of 2]**
> >
> > >**Wouldn't the general hypernymy assumptions often be violated under negation?**
> >
> > In general, it is indeed not true that there exists an implication/hypernym relation between every pair of sentences/words. For instance, as the reviewer rightfully pointed out, negation can alter the existing relations between a pair of words or sentences. However, our proposed representation quantifies how strongly the particular relation holds via the distance $d(M_u \land M_v, M_v)$. Hence, although we used this assumption to test whether our representations can capture asymmetric semantic relations, our definitions remain applicable more broadly.

---

> > ### Comment · Reviewer_7D5R · 2023-11-22
> >
> > I thank the authors for the extensive reply! I'll maintain my score, but I was on the fence, and would have raised my score slightly to a 7 if that was an option.  Some concerns were addressed, but I still have worries both about the computationally intensive/pairwise nature of the work, and I am still not convinced that their assertion that "Our representations can actually be used in much the same way as traditional embedding models such as CLIP or BERT" actually holds true in practice.

---

### Official Review · Reviewer_ecJy · 2023-10-31

**Soundness:** 3 good
**Presentation:** 3 good
**Contribution:** 3 good
**Rating:** 8
**Confidence:** 4

**Summary:**

The paper proposes a new method of sentence or phrase similarity measures using Decoder-only language models.  Decoder-only language models generate a continual string of tokens given an input. The proposed method measures two given sentences as individual inputs to a decoder-only model and measures the distributional similarity between the probabilities of multiple possible continual strings, named trajectories, for both of the inputs.
While it cannot catch up with the recent contrastive learning-based sentence similarity models, it outperforms most off-the-shelf encoder-based sentence representation models.
Although there are limitations especially high computational costs, the proposed method is interesting and unique.

**Strengths:**

- The paper is overall well-written. I had no difficulty in reading and understanding this paper.
- This paper presents an interesting and unique usage of decoder-only language models for measuring sentence similarity.
- The proposed method shows better performance on sentence similarity tasks over encoder-based baselines.

**Weaknesses:**

- For the baseline encoder-only models, it is better to include larger models like BERT-large and RoBERTa-large using their CLS tokens and token averages.
- The discussion about partial ordering between sentences is a bit puzzling. Since Tu and Tv are samples from u and v respectively, the former set of trajectories usually gives high Mu values for u and vise versa, so it hardly happens that Mu < Mv or Mu > Mv for all t in Tu U Tv. Besides, the discussion of entailment suddenly shifts from the comparison between Mu and Mv to the comparison between d(Mu cup Mv, Mu) and d(Mu cup Mv, Mv). This part is also quite puzzling.
- For the experiments of entailment and hypernym/hyponym evaluation, there is an assumption that either of those relations exists between the given input pairs, which is not realistic.
- As is pointed out in the limitation paragraph, the proposed model is computationally higher than other baseline models. In the Appendix, it is tested using a fixed set of trajectories, which seems to cause a big performance degradation.

**Questions:**

- For the Hyponym test, the contexts for a given word u are not sampled but retrieved from. It is not clear how the value Mu(s u t) is obtained.
- Is it guaranteed that the set of trajectories for an input u always the same? Or, does it vary each time sampling for a sentence u is conducted?
- The Autoregressive Model Baselines look very weak. I am not sure they are worth being included.

---

> ### Author Response · Authors · 2023-11-15
> **Response to Reviewer ecJy**
>
> We thank the reviewer for their valuable feedback and suggestions. We would like to provide a detailed response to each comment.
>
> >**For the baseline encoder-only models, it is better to include larger models like BERT-large and RoBERTa-large using their CLS tokens and token averages.**
>
> Thank you for the suggestion, we added the suggested comparisons to Table 1.
>
> >**... hardly happens that Mu < Mv or Mu > Mv for all t in Tu U Tv. Besides, the discussion of entailment suddenly shifts from the comparison between Mu and Mv to the comparison between d(Mu cup Mv, Mu) and d(Mu cup Mv, Mv).**
>
> Indeed $M_u \leq M_v$ or $M_v \leq M_u$ rarely happens in practice. This is precisely the reason we employ the alternate formulation pointed out by the reviewer. Note that $M_u \leq M_v$ can also be expressed as $d(M_u \land M_v, M_u) = 0$, where $M_u \land M_v := \min(M_u, M_v)$. Intuitively, this means that $M_u \leq M_v$ if and only if $M_u$ is fully captured by their intersection. The difference now is that this expression provides a continuous measure of how strongly the containment relation holds. We added some discussion in the main body of the paper and in Appendix F which we hope can help better clarify this. This also relates to the next comment by the reviewer:
>
> >**For the experiments of entailment and hypernym/hyponym evaluation, there is an assumption that either of those relations exists between the given input pairs, which is not realistic.**
>
> We chose to assume that a entailment/hypernym relation exists and to formulate the task as prediction of the direction of that relation in order to present a simple quantitative evaluation of our definitions. Indeed, we are not aware of datasets that provide human-labeled annotations quantifying how strongly these relations hold between two arbitrary inputs. In general, our method actually produces a single number that quantifies how strongly each relation holds (even if neither strictly holds), measured by the distance described above.
>
> >**As is pointed out in the limitation paragraph, the proposed model is computationally higher than other baseline models.**
>
> While computational cost is indeed a limitation of our method, it can be effectively mitigated by employing fewer and shorter trajectories. As evident in Figure 3 of Appendix A.1, performance starts to saturate with just 10 trajectories of maximum 10 tokens each.
>
> >**In the Appendix, it is tested using a fixed set of trajectories, which seems to cause a big performance degradation.**
>
> Since there exist no effective method of extracting zero-shot representations from autoregressive models, baselines consistently exhibit significantly inferior performance compared to our method. Therefore, despite performance degradation due to using a fixed set of trajectories, we still outperform them substantially while achieving large speed-ups.
>
> >**For the Hyponym test, the contexts for a given word u are not sampled but retrieved from. It is not clear how the value Mu(s u t) is obtained.**
>
> We elaborated on this in Appendix C.2, Algorithm 2 of the Appendix. In particular compared to sentence-level meaning, there are two main differences: (1) as correctly pointed out, contexts are retrieved rather than sampled, since “reverse-sampling” the prefixes for any given word is not trivial, and (2) the scores for each context are computed over the entire context, rather than only over the continuations following the word.
>
> >**Is it guaranteed that the set of trajectories for an input u always the same? Or, does it vary each time sampling for a sentence u is conducted?**
>
> This depends on the sampling method. Beam search generates a fixed set of trajectories, but multinomial sampling, which we use in our experiments, will generate different trajectories based on the random seed. However, we experimentally found that this variance in trajectories sampled across different seeds barely affects results.
>
> >**The Autoregressive Model Baselines look very weak. I am not sure they are worth being included.**
>
> It is not immediately obvious that those baselines do not perform well in a zero-shot setting. Most of the baseline methods are also used as components by other works such as SGPT and Sentence-T5, in both the zero-shot and fine-tuned settings. As such, even though they perform poorly, we believe that a comparison with those baselines is helpful for completeness and to be consistent with relevant prior work.

---

> ### Comment · Reviewer_ecJy · 2023-11-22
> **Thank you for your clarification responses**
>
> As most of my concerns and questions were clearly answered, I decided to upgrade my rating.

---

### Official Review · Reviewer_Lfxp · 2023-11-02

**Soundness:** 3 good
**Presentation:** 4 excellent
**Contribution:** 3 good
**Rating:** 6
**Confidence:** 4

**Summary:**

The paper presents an innovative approach to interpret and represent the meaning of text in autoregressive LLMs through the concept of trajectories -- distributions of potential text continuations. This method diverges from traditional vector space embeddings, offering a a semantic interpretation that aligns with the actual use and context of language as understood by LLMs. It effectively overcomes the challenges posed by other methods, such as prompt dependency and the need for fine-tuning, providing a more faithful reflection of the model's internal representations without additional data or model modifications.

Empirical results demonstrate that this trajectory-based approach can successfully capture complex linguistic relationships and perform competitively on semantic textual similarity tasks without any fine-tuning or prompts. Furthermore, the paper extends this approach to multimodal models, where it outperforms established benchmarks like CLIP embeddings in understanding semantic similarities across images. The main contributions of the study include a new interpretable semantic representation for autoregressive models, the alignment of these representations with human linguistic understanding, and the applicability of the method to multimodal contexts.

**Strengths:**

1. The paper's trajectory-based method for understanding LLMs is original, diverging from typical vector space models and prompt-based approaches, and introducing a new angle to distributional semantics.

2. The approach has been empirically tested against established benchmarks, indicating robust methodology and results that surpass existing techniques like CLIP embeddings on image-image similarity tasks.

3. The paper is clearly articulated, systematically presenting the new method and its implications, with illustrative examples that enhance comprehension of the proposed concepts.

4. This work is significant for its practical application in making LLMs more interpretable without extra training, it is an approach that is original as far as I know.

**Weaknesses:**

My main criticism of the paper is in the results for semantic similarity, they are far below those of contrastive methods (like 10 pts or so). I also the Sentence-T5 results are a bit misleading, that is the case without any fine-tuning. This isn't explicitly stated. There are other approaches that achieve far higher results on these tasks that do not use the training data for these tasks. I think the Sentence-T5 results in this paper are actually worse than a random encoder where random word embeddings are average together.

There are also a few typos:

Section 3: “Speicifically”
Appendix E: “presenedt”

**Questions:**

How does this model perform relative to more interpretable baselines - like random word embeddings?

---

> ### Author Response · Authors · 2023-11-15
> **Response to Reviewer Lfxp**
>
> We thank the reviewer for their constructive feedback and suggestions. We respond to each comment below.
>
> >**My main criticism of the paper is in the results for semantic similarity, they are far below those of contrastive methods (like 10 pts or so)**
>
> All of our results are obtained in the prompt-free setting,  however, if desired, state-of-the-art results among zero-shot methods with autoregressive models can be readily obtained by combining our method with prompt-engineering (see Table 7). On compositional knowledge benchmarks such as SICK-R (Table 1) and the multi-modality CxC benchmark (Table 2), the performance of our method even surpasses that of contrastive methods. We refer the reviewer to our overall comments for further discussion.
>
> > **I also (think) the Sentence-T5 results are a bit misleading, that is the case without any fine-tuning. This isn't explicitly stated.**
>
> For fair comparison, we evaluate all methods in a zero-shot manner as mentioned in Table 1. As suggested, we made this more explicit in Table 1 (Encoder-based Models -> Zero-shot Encoder-based Models), thank you.
>
> > **There are other approaches that achieve far higher results on these tasks that do not use the training data for these tasks. I think the Sentence-T5 results in this paper are actually worse than a random encoder where random word embeddings are average together.**
>
> > **How does this model perform relative to more interpretable baselines - like random word embeddings?**
>
> With regard to random-encoder results, we believe the reviewer is referring to [1], but the embeddings considered in that work for STS tasks are actually evaluated via logistic regression (with trained parameters) rather than in a zero-shot manner. As such, the results are not directly comparable with our work nor with the cited prior works (e.g., Sentence-T5 does not use logistic regression, but instead directly computes cosine similarity).
>
> We welcome any suggestions for additional comparisons or references that the reviewer believes we might have overlooked.
>
> [1] No Training Required: Exploring Random Encoders for Sentence Classification, ICLR 2019.
>
> > **Section 3: “Speicifically” Appendix E: “presenedt”**
>
> Corrected, thank you!

---

### Public Comment · ~Juri_Opitz2 · 2023-11-11
**Interesting work, two remarks/pointers**

Dear authors,

interesting paper! I have two pointers/remarks that perhaps could help strengthening the work further:

i) A pointer for the related work/motivation: We also wrote a paper on meaning-representation based interpretable text embeddings that uses a *compositional semantic space* to compute similarity, albeit in a different way than your work [1]. It could be interesting to discuss some differences in the concepts.

ii) A remark on the SNLI experiment setup. As also pointed out by R2, many people are not so much interested in the direction of entailment, but whether there is an entailment in the first place. For testing entailment classification, you could compare against our *symbolic* graph-based meaning representation approach by using the AUC score (so you do not need to tune a decision threshold) [2]. I think that such an experiment would be easy to run and would be an interesting addition to the paper (and also help address the point of R2).

[1] *SBERT studies Meaning Representations: Decomposing Sentence Embeddings into Explainable Semantic Features* https://arxiv.org/abs/2206.07023

[2] *AMR4NLI: Interpretable and robust NLI measures from semantic graphs* https://arxiv.org/abs/2306.00936

---

> ### Author Response · Authors · 2023-11-16
> **Response to Juri Opitz**
>
> Hi Juri,
>
> Thank you for your interest in our work, and for your insightful feedback and suggestions.
>
> >**We also wrote a paper on meaning-representation based interpretable text embeddings that uses a compositional semantic space to compute similarity, albeit in a different way than your work [1]. It could be interesting to discuss some differences in the concepts.**
>
> Thank you for the interesting pointer. [1] differs from our work since it focuses on fine-tuning Sentence-BERT, which is itself fine-tuned from BERT to be a sentence embedding model, to improve the interpretability of the output embeddings. In contrast, our work aims to extract meaning representations in a zero-shot manner from autoregressive models. However, the reference is indeed relevant and we have uploaded our latest revision to include it in Sec 2, Related Work.
>
> >**...many people are not so much interested in the direction of entailment, but whether there is an entailment in the first place. For testing entailment classification, you could compare against our symbolic graph-based meaning representation approach by using the AUC score (so you do not need to tune a decision threshold) [2].**
>
> AUC evaluates the probability that $(P, H, \text{entailed}) > (P’, H’, \text{non-entailed})$. This is essentially the same metric captured by direction of entailment, except that instead of using an arbitrarily sampled negative pair (pairs between which there exists neutral relation/contradiction), we construct the negative pair by swapping the premise and hypothesis.
>
> The reason we chose the (harder) direction of entailment task is to fully evaluate whether our definitions are able to capture asymmetric relations. In the case of the binary classification problem defined in [2], comparing positive pairs with entailment relations and negative pairs with (contradiction/neutral) relations does not require an asymmetric distance, and could be accomplished simply via similarity, a symmetric distance (indeed, [2] evaluates both asymmetric and symmetric distances).
>
> >**Maybe can you explain why the method cannot be used for basic entailment prediction? If it can predict the weird(er) task of strength and direction of entailment, why shouldn't it predict entailment?**
>
> Prediction of entailment is definitely possible with our method, and indeed one could follow a similar evaluation procedure as that of [2] to avoid tuning a decision threshold. However, as argued above, that kind of evaluation would not truly measure the ability of our representation to capture asymmetric relations in meaning. For this reason, we believe that the task of predicting the direction of entailment was better suited for our purposes. Investigating the effectiveness of our representation on other general semantic tasks like the one suggested is a topic that we hope to explore in future work.

---

> ### Public Comment · ~Juri_Opitz2 · 2023-11-16
> **Appreciate the answer, see your point**
>
> Thanks, much appreciate your answers and considering our work.
>
> Now I understand your point better. I wouldn't 100% agree with the statement that the direction task using only positive pairs is  "harder" than entailment. Also when you have positive and negative pairs, yes, similarity may be efficient, you are right, but I'd still argue you'd need asymmetry (H entailed by P doesn't mean P entailed by H), also the hypothesis may not be much similar to the premise, but still entailed (positive), or it can be very similar, but it may be neutral (negative).
>
> That said, I concede that in entailment you could say that similarity and asymmetry are too much conflated so I see why you tested your method in such a way that you did, to focus on the asymmetry. Just suggesting that experiments on entailment could be a nice (additional) selling point to the NLP community, where the task is well established.
>
> Anyways, I'm quite excited about your method/paper, good luck.

---

### Author Response · Authors · 2023-11-15
**Updated Rebuttal Revision + New Experiments**

We appreciate the reviewers' valuable feedback and suggestions. We emphasize that our primary objective is not to achieve state-of-the-art results on existing benchmarks. Instead, our contribution lies in proposing a working definition of meaning for autoregressive models, which can be directly applied to various downstream tasks, such as evaluating semantic alignment with human annotators. Nevertheless, obtaining state-of-the-art results among zero-shot methods (including prompt-based ones) is certainly feasible with our method. We show in Table 7 that this can be easily achieved by adding prompts to condition the set of sampled trajectories. This kind of approach, however, deviates from our original goal of defining meaning in a prompt-free manner. Therefore, these results are included in the Appendix.

As requested, we have also expanded our evaluation to more tasks, in particular those involving compositional knowledge. We show that our definition achieves state-of-the-art results across all zero-shot methods, including contrastive trained models, on the SICK-R (Sentences Involving Compositional Knowledge) dataset.

We have uploaded a revised draft with the following changes:
- Added results on SICK-R (Sentences Involving Compositional Knowledge), where we achieve best performance across all methods, including contrastive-trained models (Tables 1,4,6,7 and Figure 4)
- Added results on STS and SICK-R using prompts, where we achieve state-of-the-art results across all zero-shot methods on autoregressive architectures, including those based on prompt engineering (Table 7)
- Added comparison to BERT-large and RoBERTa-large (Table 1)
- Revised our discussion on partial orderings in the paragraph “Containments of semantic representations”

---

> ### Public Comment · ~Juri_Opitz2 · 2023-11-15
> **Any update on entailment? [Resolved, thank you]**
>
> ~~Hi,~~
>
> ~~it's not clear to me why the method could not be used for predicting entailment, a task that is much more accepted and interesting than "strength of relation". If you decide to stick with "strength of relation", you may at least consider the multiple-annotator labels that are contained in the NLI data sets, otherwise this task is not well defined.~~
>
> ~~Maybe can you explain why the method cannot be used for basic entailment prediction? If it can predict the weird(er) task of strength and direction of entailment, why shouldn't it predict entailment? Appreciate any response.~~

---

> > ### Author Response · Authors · 2023-11-16
> > **Please see "Response to Juri Opitz"**
> >
> > We refer to the response below, where we address this along with the earlier comments.

---

### Meta-Review · Area_Chair_xxuw · 2023-12-13

**Metareview:**

This paper presents a novel method for measuring semantic textual similarity. Instead of using an encoder to produce vector embeddings, the paper uses a autoregressive LM to sample multiple continuations of each input. The likelihoods of these continuations conditioned on both inputs are used to create a similarity score. The method outperforms zero-shot baselines for semantic textual similarity. Reviews were all positive, praising the novelty of the proposed method, the competitive results with other encoder-based baselines, and the clarity of writing in the paper itself. The common weakness brought up by reviewers is the high computational costs of the proposed approach.

**Justification For Why Not Higher Score:**

The proposed approach is computationally expensive -- while it represents an interesting new approach, to have broader practical impact, further improvements to efficiency could be considered.

**Justification For Why Not Lower Score:**

The method is refreshingly novel and outperforms zero-shot baselines.

---

### Decision · Program_Chairs · 2024-01-16

Accept (poster)